# FLAIM: AIM-BASED SYNTHETIC DATA GENERATION IN THE FEDERATED SETTING

## ABSTRACT

Preserving individual privacy while enabling collaborative data sharing is crucial for organizations. Synthetic data generation is one solution, producing artificial data that mirrors the statistical properties of private data. While numerous techniques have been devised under differential privacy, they predominantly assume data is centralized. However, data is often distributed across multiple clients in a federated manner. In this work, we initiate the study of federated synthetic tabular data generation. Building upon a SOTA central method known as AIM, we present *DistAIM* and *FLAIM*. We show it is straightforward to distribute AIM, extending a recent approach based on secure multi-party computation which necessitates additional overhead, making it less suited to federated scenarios. We then demonstrate that naively federating AIM can lead to substantial degradation in utility under the presence of heterogeneity. To mitigate both issues, we propose an augmented FLAIM approach that maintains a private proxy of heterogeneity. We simulate our methods across a range of benchmark datasets under different degrees of heterogeneity and show this can improve utility while reducing overhead.

## 1 INTRODUCTION

Modern computational applications are predicated on the availability of significant volumes of high-quality data. Increasingly, such data is not freely available: it may not be collected in the volume needed, and may be subject to privacy concerns. Recent regulations such as the General Data Protection Regulation (GDPR) restrict the extent to which data collected for a specific purpose may be processed for some other goal. The aim of *synthetic data generation* (SDG) is to solve this problem by allowing the creation of realistic artificial data that shares the same structure and statistical properties as the original data source. SDG is an active area of research, offering the potential for organisations to share useful datasets while protecting the privacy of individuals. (Assefa et al., 2020; Mendelevitch and Lesh, 2021; van Breugel and van der Schaar, 2023).

SDG methods fall into two main categories: deep learning (Kingma et al., 2019; Xu et al., 2019; Goodfellow et al., 2020) and statistical models, which are popular for tabular data (Young et al., 2009; Zhang et al., 2017). Nevertheless, without strict privacy measures in place, it is possible for SDG models to leak information about the data it was trained on (Murakonda et al., 2021; Stadler et al., 2022; Houssiau et al., 2022). It is common for deep learning approaches such as Generative Adversarial Networks (GANs) to produce verbatim copies of training examples which leaks privacy (Srivastava et al., 2017; van Breugel et al., 2023). A standard approach to prevent leakage is to use Differential Privacy (DP) (Dwork and Roth, 2014). DP is a formal definition which ensures the output of an algorithm does not depend heavily on any one individual's data by introducing calibrated random noise. Under DP, statistical models have become SOTA for tabular data and commonly outperform deep learning counterparts (Tao et al., 2021; Liu et al., 2022). Approaches are based on Bayesian networks (Zhang et al., 2017), Markov random fields (MRF) (McKenna et al., 2019) and iterative marginal-based methods (Liu et al., 2021; Aydore et al., 2021; McKenna et al., 2022).

Private SDG methods perform well in centralized settings where a trusted curator holds all the data. However, in many settings, data cannot be easily centralized. Instead, there are multiple participants each holding a small private dataset who wish to generate synthetic data. Federated learning (FL) is a paradigm that applies when multiple parties want to collaboratively train a model without sharing data directly (Kairouz et al., 2019). In FL, local data remains on-device, and only model updates

are transmitted back to a central aggregator (McMahan et al., 2017b). FL methods commonly adopt differential privacy to provide formal privacy guarantees and is widely used in deep learning (McMahan et al., 2017a; Kairouz et al., 2021; Huba et al., 2022). However, there has been minimal focus on federated SDG: we only identify a recent effort to distribute Multiplicative Weights with Exponential Mechanism (MWEM) via secure multi-party computation (SMC) (Pereira et al., 2022).

In this work, we study generating differentially private tabular data in the practical federated scenario where only a small proportion of clients are available per-round and clients experience strong data heterogeneity. We propose FLAIM, a variation of the current SOTA central DP algorithm AIM (McKenna et al., 2022). We show how an analog to traditional FL training can be formed for AIM with clients performing a number of local steps before sending model updates to the server in the form of noisy marginals. We highlight how this naive extension can suffer severely under strong heterogeneity which is exacerbated when only a few clients participate per round. To circumvent this, we modify FLAIM by extending components of central AIM to the federated setting, such as augmenting clients' local choices via a private proxy for heterogeneity to ensure decisions are not adversely affected by heterogeneity.

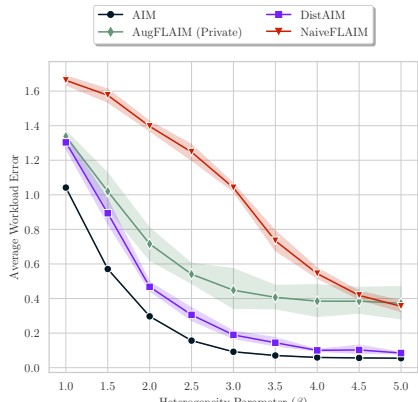

Figure 1: We construct a synthetic dataset denoted SynthFS that exhibits feature skew and plot the average error over a workload of 64 3-way marginals whilst varying heterogeneity, $\varepsilon = 3$.

**Example.** *Figure 1 presents a federated scenario where 10% of 100 clients participate per round. Each client holds data with varying degrees of feature skew, where a larger $\beta$ implies less heterogeneity (see Section 5). We use four variations of AIM: Central AIM, Distributed AIM (a modification of Pereira et al. (2022), replacing MWEM with AIM), a naive FLAIM approach and a FLAIM variation which augments local decisions to counteract skew (AugFLAIM). We plot the L1 error over a workload of 3-way marginal queries. Under the presence of client subsampling there is an inevitable gap between central and distributed AIM. By naively federating AIM, client decisions made during local training are strongly affected by heterogeneity while distributed AIM is not, resulting in a large loss in utility. This utility gap is almost closed in high skew scenarios by penalising clients' local decisions via a private measure of heterogeneity (AugFLAIM).*

Our main contributions are as follows:

- We initiate a study of iterative marginal-based methods in the federated setting with strong data heterogeneity and client subsampling. We show it is possible to extend the recent work of Pereira et al. (2022) to form a distributed version of AIM (DistAIM).

- We propose FLAIM, a variation of the AIM algorithm (McKenna et al., 2022) that is more suited to federated scenarios than DistAIM. We propose extensions based on augmenting the utility scores used in AIM decisions via a private proxy that reduces the effect heterogeneity has on local decisions, often resulting in increased model performance.

- We perform an extensive empirical study into the effect of heterogeneity on federating AIM. We show that FLAIM often results in utility close to DistAIM but reduces the need for heavyweight SMC techniques, resulting in reduced overhead. Our open-source code is available at https://anonymous.4open.science/r/flaim-0151/.

## 2 PRELIMINARIES

We assume the existence of $K$ participants each holding local datasets $D_1, \ldots, D_K$ over a set of $d$ attributes such that the full dataset is denoted $D := \cup_k D_k$. Additionally, we assume that each attribute is categorical[1]. For a record $\boldsymbol{x} := (x_1, \ldots, x_d) \in D$ we denote $x_i$ as the value of attribute $i$. For each attribute $i \in [d]$, we define $A_i$ as the set of discrete values that $x_i$ can take. For a subset of attributes $q \subseteq [d]$ we abuse notation and let $x_q$ be the subset of $\boldsymbol{x}$ with attributes in the set $q$. We

---

[1]As in prior work, we discretize continuous features via uniform binning. See Appendix B.1 for more details.

are mostly concerned with computing marginal queries over $D$ (or individual $D_k$). Let $q \subseteq [d]$ and define $A_q := \prod_{i \in q} A_i$, as the set of values $q$ can take and $n_q := |A_q|$ as the cardinality of $q$.

**Definition 2.1** (Marginal Query). *A marginal query for $q \subseteq [d]$ is a function $M_q : \mathcal{D} \to \mathbb{R}^{n_q}$ s.t. each entry is a count of the form $(M_q(D))_j := \sum_{x \in D} \mathbf{1}[x_q = a_j]$, $\forall j \in [n_q], a_j \in A_q$.*

The goal in workload-based synthetic data generation is to generate a synthetic dataset $\hat{D}$ that minimises $\mathrm{Err}(D, \hat{D})$ over a given workload of (marginal) queries $Q$. We follow existing work (McKenna et al., 2022) and study the average workload error under the $L_1$ norm.

**Definition 2.2** (Average Workload Error). *Denote the workload $Q = \{q_1, \ldots, q_m\}$ as a set of marginal queries where each $q \subseteq [d]$. The average workload error for synthetic dataset $\hat{D}$ is defined $\mathrm{Err}(D, \hat{D}; Q) := \frac{1}{|Q|} \sum_{q \in Q} \|M_q(D) - M_q(\hat{D})\|_1$*

We are interested in producing a synthetic dataset $\hat{D}$ with marginals close to that of $D$. However, in the federated setting it is often impossible to form the global dataset $D := \cup_k D_k$ due to privacy restrictions or client availability. Instead the goal is to learn sufficient information from local datasets $D_k$ and train a model that learns $M_q(D)$. For any $D_k$, the marginal query $M_q(D_k)$ and local workload error $\mathrm{Err}(D_k, \hat{D})$ are defined analogously.

**Differential Privacy (DP)** (Dwork et al., 2006) is a formal notion that guarantees the output of an algorithm does not depend heavily on any individual. We seek to guarantee $(\varepsilon, \delta)$-DP, where the parameter $\varepsilon$ is called the *privacy budget* and determines an upper bound on the privacy leakage of the algorithm. The parameter $\delta$ defines the probability of failing to meet this, and is set very small. DP has many properties including composition, meaning that if two algorithms are $(\varepsilon_1, \delta_1)$-DP and $(\varepsilon_2, \delta_2)$-DP respectively, then the joint output of both algorithms satisfies $(\varepsilon_1 + \varepsilon_2, \delta_1 + \delta_2)$-DP. Tighter bounds are obtained with zero-Concentrated DP (zCDP) (Bun and Steinke, 2016):

**Definition 2.3** ($\rho$-zCDP). *A mechanism $\mathcal{M}$ is $\rho$-zCDP if for any two neighbouring datasets $D, D'$ and all $\alpha \in (1, \infty)$ we have $D_\alpha(\mathcal{M}(D)|\mathcal{M}(D') \le \rho \cdot \alpha$, where $D_\alpha$ is Renyi divergence of order $\alpha$.*

One can convert $\rho$-zCDP to obtain an $(\varepsilon, \delta)$-DP guarantee. The notion of "adjacent" datasets can lead to different privacy definitions. We assume *example-level privacy*, which defines two datasets $D$, $D'$ to be adjacent if $D'$ can be formed from the addition/removal of a single individual from $D$. To satisfy DP it is common to require bounded *sensitivity* of the function we wish to privatize.

**Definition 2.4** (Sensitivity). *Let $f : \mathcal{D} \to \mathbb{R}^d$ be a function over a dataset. The $L_2$ sensitivity of $f$ denoted $\Delta_2(f)$ is defined as $\Delta_2(f) := \max_{D \sim D'} \|f(D) - f(D')\|_2$, where $D \sim D'$ represents the example-level relation between datasets. Similarly, $\Delta_1(f)$ is defined with the $L_1$ norm.*

We focus on two building block DP mechanisms that are core to existing DP-SDG algorithms. These are the Gaussian and Exponential mechanisms (Dwork and Roth, 2014).

**Definition 2.5** (Gaussian Mechanism). *Let $f : \mathcal{D} \to \mathbb{R}^d$, the Gaussian mechanism is defined as $GM(f) = f(D) + \Delta_2(f) \cdot \mathcal{N}(0, \sigma^2 I_d)$. The Gaussian mechanism satisfies $\frac{1}{2\sigma^2}$-zCDP.*

If we wish to normalize the marginal $M_q(D)$ the sensitivity is at most $1/|D|$ since adding or removing a single example contributes $1/|D|$ to a single entry.

**Definition 2.6** (Exponential Mechanism). *Let $u(q; \cdot) : \mathcal{D} \to \mathbb{R}$ be a utility function defined for all candidates $q \in Q$. The exponential mechanism releases $q$ with probability $\mathbb{P}[\mathcal{M}(D) = q] \propto \exp(\frac{\varepsilon}{2\Delta} \cdot u(q; D))$, with $\Delta := \max_q \Delta_1(u(q; D))$. This satisfies $\frac{\varepsilon^2}{8}$-zCDP.*

**Iterative Methods (Select-Measure-Generate).** Recent SOTA methods for private tabular data generation follow the "Select-Measure-Generate" paradigm which is the core focus of our work. These are broadly known as iterative methods (Liu et al., 2021) and usually involve training a graphical model via noisy marginals over a number of steps. In this work, we focus on AIM (McKenna et al., 2022), an extension of the classical MWEM algorithm (Hardt et al., 2012), which replaces the multiplicative weight update with a graphical model inference procedure called Private-PGM (McKenna et al., 2019). PGM learns a Markov random field (MRF) and applies post-processing optimisation to ensure a level of consistency in the generated data. PGM can answer queries without directly generating data from the model, thus avoiding additional sampling errors.

In outline, given a workload of queries $Q$, AIM proceeds as follows (full details in Appendix A.1): At each round, via the exponential mechanism, **select** the query that is worst-approximated by the current synthetic dataset. Under the Gaussian mechanism **measure** the chosen marginal and update the graphical model via PGM. At any point, we can **generate** synthetic data via PGM that best explains the observed measurements. AIM begins round $t$ by computing utility scores for each query $q \in Q$ of the form $u(q; D) = w_q \cdot (\|M_q(D) - M_q(\hat{D}^{(t-1)})\|_1 - \sqrt{\frac{2}{\pi}} \cdot \sigma_t \cdot n_q)$, where $\hat{D}^{(t-1)}$ is the current PGM model. The core idea is to select marginals that are high in error (first term) balanced with the expected error from measuring the query under Gaussian noise with variance $\sigma_t^2$ (second term). The utility scores are weighted by $w_q := \sum_{r \in Q} |r \cap q|$ which calculates the overlap of other marginals in the workload with $q$. The sensitivity of the resulting exponential mechanism is $\Delta = \max_q w_q$ since measuring $\|M_q(D) - M_q(\hat{D}^{(t-1)})\|_1$ has sensitivity 1 which is weighted by $w_q$. Once a query is selected it is measured by the Gaussian mechanism with variance $\sigma_t^2$ and sensitivity 1. An update to the model via PGM is then applied using all observed measurements so far.

Given a workload $Q$, the goal is to learn a synthetic dataset $\hat{D}$ that best approximates $D$ over the workload e.g., $M_q(\hat{D}) \approx M_q(D), \forall q \in Q$. However, computing statistics directly from $D$ is not possible in a federated setting since clients may not be available or the entire population may be unknown. For simplicity, we assume that each participant is sampled with probability $p$ to participate in the current round. We also assume that each participant has a local dataset $D_k$ that exhibits heterogeneity. This could manifest as significant label-skew; a varying number of samples; or as data that is focused on a specific subset of feature values. We make clearer how we model heterogeneity experimentally in Section 5, with full details presented in Appendix B.2.

## 3 DISTRIBUTED AIM

Our first proposal, DistAIM, translates the AIM algorithm directly into the federated setting by having participants jointly compute each step. To do so, participants must collaborate privately and securely, such that no one participant's raw marginals are revealed to the others. The "select" and "measure" steps are the only components requiring private data, and hence we need to implement distributed DP mechanisms for these steps. We present an overview here with full details in Appendix A.2.

Pereira et al. (2022) describe one such approach for MWEM. They utilize various secure multi-party computation (SMC) primitives based on secret-sharing (Araki et al., 2016). They first assume that all participants secret-share their workload answers to compute servers. These compute servers then implement secure exponential and Laplace mechanisms over shares of marginals via standard SMC operations (Keller, 2020). This approach has two drawbacks: first, their cryptographic solution incurs both a computation and communication overhead which may be prohibitive in federated scenarios. Secondly, their approach is based on MWEM which results in a significant loss in utility. Furthermore, MWEM is memory-intensive and does not scale to high-dimensional datasets. Instead, we apply the framework of Pereira et al. (2022) to AIM, with minor modifications. Compared to AIM, the DistAIM approach has some important differences:

**Client participation:** At each round only a subset of participants are available to join the AIM round. For simplicity, we assume clients are sampled with probability $p$. In expectation, $pK$ clients contribute their secret-shared workload answers e.g., $\{[\![M_q(D_k)]\!] : q \in Q\}$ to compute servers. This is an immediate difference with the setting of Pereira et al. (2022), where it is assumed that all clients are available to secret-share marginals before training. Instead in DistAIM, the secret-shares from participants are aggregated across rounds and the "select" and "measure" steps are carried out via compute servers over the updated shares at a particular round. Compared to the central setting, DistAIM incurs some additional error due to subsampling.

**Select step:** A key difficulty in extending AIM (or MWEM) to a distributed setting is the use of the exponential mechanism (EM). In order to apply EM, the utility scores $u(q; D)$ must be calculated. Following Protocol 2 of Pereira et al. (2022), sampling from the EM can be done over secret-shares of the marginal $[\![M_q(D_k)]\!]$ since the utility scores $u(q; D)$ depend only linearly in $M_q(D_k)$.

**Measure step:** Once a marginal has been sampled, it must be measured. Protocol 3 in Pereira et al. (2022) proposes one way to securely generate Laplace noise between compute servers and aggregate this with secret-shared marginals. To remain consistent with AIM, we use Gaussian noise instead.

---

**Algorithm 1** FLAIM

---

**Input:** $K$ participants with data $D_1, \ldots, D_k$, sampling rate $p$, workload $Q$, privacy parameters $(\varepsilon, \delta)$

1: **for** each global round $t$ **do**
2:     Form round participants $P_t$ by sampling each client $k \in [K]$ with probability $p$
3:     **for** each client $k \in P_t$ **do**
4:         **for** each local step $l \in [s]$ **do**
5:             Filter the workload $Q \leftarrow Q \setminus \{|q| = 1 : q \in Q\}$ (Private only)
6:             Compute a heterogeneity measure for each $q \in Q$

$$\tilde{\tau}_k(q) \leftarrow \begin{cases} 0 & \text{Naive} \\ \tau_k(q) := \|M_q(D_k) - M_q(D)\|_1 & \text{Non-private} \\ \frac{1}{|q|} \sum_{j \in q} \|M_{\{j\}}(D_k) - M_{\{j\}}(\hat{D}^{(t-1)})\|_1 & \text{Private} \end{cases}$$

7:             **Select** $q_{t+l} \in Q$ using the exponential mechanism with utility score

$$u(q; D_k) \leftarrow w_q \left( \|M_q(D_k) - M_q(\hat{D}^{(t-1)+l})\|_1 - \sqrt{\tfrac{2}{\pi}} \cdot \sigma_{(t-1)+l} \cdot n_q - \tilde{\tau}_k(q) \right)$$

8:             **Measure** the chosen marginal $\tilde{M}_q(D_k) := M_q(D_k) + \mathcal{N}\left(0, \sigma^2_{(t-1)+l}I\right)$
9:             **Estimate** the new local model via PGM ($s > 1$ only)

$$\hat{D}_k^{(t-1)+l} \leftarrow \arg\min_{p \in \mathcal{S}} \sum_{i=1}^{(t-1)+l} \tfrac{1}{\sigma_i} \|M_{q_i}(p) - \tilde{M}_{q_i}(D_k)\|_2$$

10:         Share all 1-way marginals under SecAgg, $\mathcal{M}_k^1 \leftarrow \{(t, j, [\![M_{\{j\}}(D_k)]\!]\}_{j \in [d]}$ (Private only)
11:         Send $\mathcal{M}_k \leftarrow \{(t, q_{t+l}, [\![M_{q_{t+l}}(D_k)]\!])\}_{l \in [s]} \cup \mathcal{M}_k^1$ to the server, shared via SecAgg
12:     Server forms updated measurement list for round $t$, $\mathcal{M}^t := \cup_{k \in P_t} \mathcal{M}_k$
13:     **for** each unique $q \in \mathcal{M}^t$ aggregate associated marginals $\tilde{M}_q^t := \sum_{\{k : M_q \in \mathcal{M}_k\}} [\![M_q(D_k)]\!] + N(0, \sigma_t^2)$
14:     **for** each $\tilde{M}_q^t$ compute weight $\alpha_q^t$ (see Section 4.3) $:= \begin{cases} 1/\sigma_t, & \text{Naive} \\ N_q^t/\sigma_t, & \text{Non-private} \\ \tilde{N}_q^t/\sigma_t, & \text{Private} \end{cases}$
15:     Server updates measurement list $\mathcal{M}$ with each $(t, q, \tilde{M}_q^t, \alpha_q^t)$ and updates the global model

$$\hat{D}_t \leftarrow \arg\min_{p \in \mathcal{S}} \sum_{(t, q, \tilde{M}_q^t, \alpha_q^t) \in \mathcal{M}} \alpha_q^t \|M_q(p) - \tilde{M}_q^t\|_2$$

---

**Estimate step:** Under the post-processing properties of differential privacy the compute server(s) are free to use the noisy marginals with PGM to update the graphical model, as in the centralized case.

## 4 FLAIM: A FL ANALOG FOR AIM

While DistAIM is one solution to federated SDG, it is not defined within the standard FL framework where clients typically perform a number of local steps before sending model updates to a server. Furthermore, it's SMC-based approach can result in large overheads. This leads us to design an AIM approach that is analogous to traditional Federated Learning (FL), where only lightweight SMC is needed in the form of secure-aggregation (Bell et al., 2020). In FL, the standard approach for training models is to do more computation on-device, and have clients perform multiple local steps before sending a model update. The server aggregates all clients' updates and performs a step to the global model (McMahan et al., 2017b). When combined with DP, model updates are aggregated via secure-aggregation schemes and noise is added either by a trusted server or in a distributed manner. In the case of AIM, we denote our analogous FL approach as FLAIM. In FLAIM, the selection step of AIM is performed locally by clients. The server aggregates measurements chosen by clients during local training and noise is added by the server.

FLAIM is outlined in Algorithm 1. We present three variations, with differences highlighted in color. Shared between all variations are the key differences with DistAIM displayed in blue. First, NaiveFLAIM, a straightforward translation of AIM into the federated setting with no modifications. In Section 4.1, we explain the shortcomings of such an approach which stems from scenarios where clients' local data exhibits strong heterogeneity. Motivated by this, Section 4.2 proposes AugFLAIM

(Non-Private) a variant of FLAIM that uses heterogeneity as a measure to augment local utility scores. This quantity is non-private and not obtainable in practice, but provides an idealized baseline. Lastly, Section 4.3 introduces AugFLAIM (Private), which again augments local utility scores but with a private proxy of heterogeneity alongside other heuristics to improve utility.

All FLAIM variants proceed by sampling clients to participate in round $t$. Each client performs a number of local steps $s$, which consist of performing a local selection step using the exponential mechanism, measuring the chosen marginal under local noise and updating their local model via PGM. When each client finishes local training, they send back each chosen query $q$ alongside the associated marginal $[\![M_q(D_k)]\!]$, secret-shared via secure-aggregation. These marginals are aggregated and noise is added by the central server. Hence, the local training is done under local differential privacy (LDP) so as to not leak any privacy between steps, whereas the resulting global update is done under a form of distributed privacy where noise is added by the central server to the securely-aggregated marginals. We assume all AIM methods run for $T$ global rounds. AIM is also proposed with budget annealing that sets $T$ adaptively and we explore this in our experiments (see Appendix B.4 for details).

## 4.1 ISSUES WITH NAIVEFLAIM UNDER HETEROGENEITY

In federated settings, participants often exhibit strong heterogeneity in their local datasets. That is, clients' local datasets $D_k$ can differ significantly from the global dataset $D := \cup_k D_k$. Such heterogeneity will affect AIM in both the "select" and "measure" steps. If $D_k$ and $D$ are significantly different then the local marginal $M_q(D_k)$ will differ from the true marginal $M_q(D)$. We quantify heterogeneity for a client $k$ and query $q \in Q$ via the $L_1$ distance $\tau_k(q) := \|M_q(D_k) - M_q(D)\|_1$.

In FLAIM, we proceed by having all participating clients perform a number of local steps. The first stage involves carrying out a local "select" step based on utility scores of the form $u(q; D_k) \propto \|M_q(D_k) - M_q(\hat{D}^{(t-1)})\|_1$. Suppose for a particular client $k$ there exists a $q \in Q$ such that $M_q(D_k)$ exhibits strong heterogeneity. If at step $t$ the current model $\hat{D}^{(t-1)}$ is a good approximation of $D$ for query $q$, then it is likely that client $k$ ends up selecting any query that has high heterogeneity since $u(q; D_k) \propto \tau_k(q)$. This mismatch can harm model performance and is compounded by having many clients select and measure (possibly differing) marginals that are likely to be skewed and so the model can be updated in a way that drifts significantly from $D$.

## 4.2 AUGFLAIM (NON-PRIVATE): CIRCUMVENTING HETEROGENEITY

The difficulty above arises as clients choose marginals via local applications of the exponential mechanism with a score that does not account for underlying heterogeneity. We have $u(q; D_k) \propto$

$$\|M_q(D_k) - M_q(\hat{D})\|_1 \leq \|M_q(D) - M_q(\hat{D})\| + \|M_q(D_k) - M_q(D)\|_1 \propto u(q; D) + \tau_k(q)$$

Hence we correct local decisions by down-weighting marginals based on $\tau_k(q)$ and construct local utility scores of the form: $u(q; D_k) \propto \|M_q(D_k) - M_q(\hat{D})\| - \tau_k(q)$ where $\tau_k(q)$ is an exact $L_1$ measure of heterogeneity for client $k$ at a marginal $q$. Unfortunately, measuring $\tau_k(q)$ under privacy constraints is not feasible. That is, $\tau_k(q)$ depends directly on $M_q(D)$, which is exactly what were are trying to learn! We denote AugFLAIM (Non-Private) as the variation of FLAIM that assumes $\tau_k(q)$ is known and augments the utility scores as above. This is an idealized baseline to compare with.

## 4.3 AUGFLAIM (PRIVATE): PROXY FOR HETEROGENEITY

Since augmenting utility scores directly via $\tau_k(q)$ is difficult, we seek a proxy $\tilde{\tau}_k(q)$ that is reasonably correlated with $\tau_k(q)$ and can be computed under privacy. This proxy measure can be used to correct local utility scores, penalising queries via $\tilde{\tau}_k(q)$. This helps ensure clients select queries that are not adversely affected by heterogeneity. We propose the following proxy

$$\tilde{\tau}_k(q) := \tfrac{1}{|q|} \sum_{j \in q} \|M_{\{j\}}(D_k) - \tilde{M}_{\{j\}}(D)\|_1$$

Instead of computing a measure for each $q \in Q$, we compute one for each feature $j \in [d]$, where $\tilde{M}_{\{j\}}(D_k)$ is a noisy estimate of the 1-way marginal for feature $j$. For a particular $q \in Q$, we average the heterogeneity of the associated features contained in $q$. Such a $\tilde{\tau}_k(q)$ relies only on estimating the distribution of each feature. This estimate can be refined across multiple federated rounds as

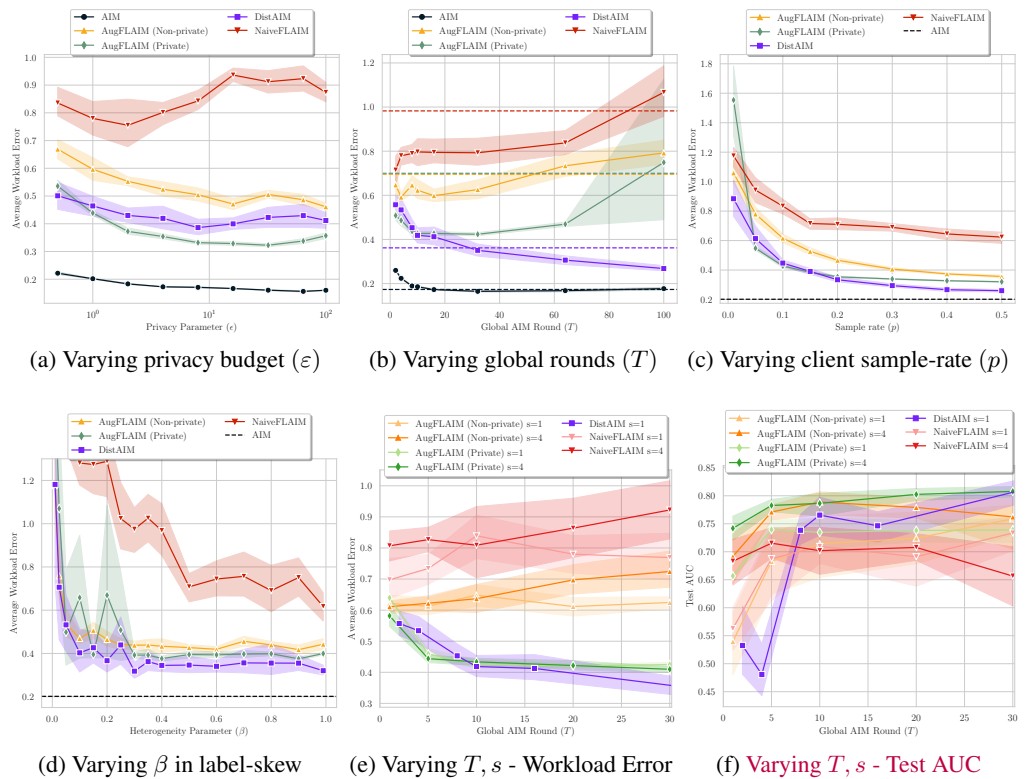

Figure 2: Varying (FL)AIM Parameters on Adult; Unless otherwise stated $p = 0.1, K = 100, \varepsilon = 1$

each participant can measure $M_{\{j\}}(D_k)$ for each $j \in [d]$ and have the server sum and add noise (via SecAgg) to produce a new private estimate $\tilde{M}_{\{j\}}(D)$ each round. We add two further enhancements:

**1. Filtering and combining 1-way marginals (Line 10).** As we require clients to estimate all feature distributions at each round, we remove 1-way marginals from the workload $Q$ to prevent clients from measuring the same marginal twice. All 1-way marginals that are estimated for heterogeneity are also fed back into PGM to improve the global model at each round.

**2. Weighting Marginals (Line 14).** In PGM, measurements are weighted by $\alpha = 1/\sigma_t$, so those that are measured with less noise have more importance in the optimisation of model parameters. Both AugFLAIM variations adopt an additional weighting scheme that includes the total sample size that contributed to a particular marginal $q$ at round $t$, $N_q^t := \sum_{\{k : M_q^t \in \mathcal{M}_k\}} |D_k|$ where the weight becomes $\alpha_q^t = N_q^t / \sigma_t$. This relies on knowing the total number of samples that are aggregated. In some cases, the size of local datasets may be deemed private. In such scenarios, it can be estimated from the noisy marginal $\tilde{M}_q$ by summing each contribution in the histogram to produce $\tilde{N}_q$.

The privacy guarantees of all FLAIM variations follow directly from those of AIM. The use of a heterogeneity measure incurs an additional sensitivity cost for the exponential mechanism and AugFLAIM (Private) incurs an additional privacy cost in measuring each of the $d$ features at every round. The following lemma captures this. See Appendix A.3 for the full proof.

**Lemma 4.1.** *For any number of global rounds $T$ and local rounds $s$, FLAIM satisfies $(\varepsilon, \delta)$-DP, under Gaussian budget allocation $r \in (0, 1)$ by computing $\rho$ according to Lemma A.2, and setting*

$$
\sigma_t = \begin{cases} \sqrt{\frac{Ts+d}{2 \cdot r \cdot \rho}}, & \textit{Naive or Non-private} \\ \sqrt{\frac{T(s+d)}{2 \cdot r \cdot \rho}}, & \textit{Private} \end{cases}, \quad \varepsilon_t = \sqrt{\frac{8 \cdot (1 - r) \cdot \rho}{Ts}}
$$

*For AugFLAIM methods, the exponential mechanism is applied with sensitivity $\Delta := \max_q 2w_q$.*

## 5 EXPERIMENTAL EVALUATION

We present an empirical comparison of approaches outlined in previous sections. We utilize benchmark tabular datasets from the UCI repository (Dua and Graff, 2017): Adult, Magic, Mushroom and Nursery. For full details see Appendix B.1. We also construct a synthetic dataset with feature-skew denoted SynthFS with the full construction detailed in Appendix B.1.1. We evaluate our methods in three ways: average workload error (as defined in Section 2), average negative log-likelihood evaluated on a holdout set and the test AUC of a decision tree model trained on synthetic data and tested on the holdout. For all datasets, we simulate heterogeneity by forming non-IID splits in one of two ways: The first is by performing dimensionality reduction and clustering nearby points to form client partitions that have strong feature-skew. We call this the "clustering" approach. For experiments that require varying heterogeneity, we form splits via an alternative label-skew method popularized by Li et al. (2022). This samples a label distribution $p_c \in [0, 1]^K$ for each class $c$ from a Dirichlet($\beta$) where larger $\beta$ results in less heterogeneity. See Appendix B.2 for full details.

In the following, all experiments have $K = 100$ clients with partitions formed from the clustering approach unless stated otherwise. We train models on a fixed workload of 3-way marginal queries chosen uniformly at random and average results over 10 independent runs. We compare central AIM and DistAIM against NaiveFLAIM and our two variants that augment local scores: AugFLAIM (Non-Private) using $\tau_k(q)$ only and AugFLAIM (Private) using proxy $\tilde{\tau}_k(q)$ with filtering and combining 1-ways. Further plots on datasets besides Adult are in Appendix C.

**Varying the privacy budget** $(\varepsilon)$. In Figure 2a, we plot the workload error whilst varying $\varepsilon$ on Adult, sampling $p = 0.1$ clients per round and set $T = 10$. First, we observe a clear gap in performance between DistAIM and central AIM due to the error from subsampling a small number of clients per round. We observe that naively federating AIM gives the worst performance even as $\varepsilon$ becomes large. Furthermore, augmenting utility scores makes a clear improvement in workload error, particularly for $\varepsilon > 1$. By estimating feature distributions at each round, AugFlaim (Private) can obtain performance that matches or sometimes improves upon DistAIM for larger values of $\varepsilon$. Further note that AugFLAIM (Private) has lower error than AugFLAIM (Non-private) which may seem counter-intuitive. However, AugFLAIM (Non-private) is still trained under DP, only the utility scores $\tau_k(q)$ are used non-privately and is trained without estimating 1-ways at every round. In Appendix C, we explore this further via an ablation which confirms this impacts utility significantly.

**Varying the number of global AIM rounds** $(T)$. In Figure 2b, we vary the number of global AIM rounds and fix $\varepsilon = 1$. Additionally, we plot the setting where $T$ is chosen adaptively by budget annealing. This is shown in dashed lines for each method. First observe with DistAIM, the workload error decreases as $T$ increases. Since compute servers aggregate secret-shares across rounds, then as $T$ grows large, most clients will have been sampled and the server(s) have workload answers over most of the (central) dataset. For all FLAIM variations, the workload error increases as $T$ increases, since they are more sensitive to the increased amount of noise that is added when $T$ is large. For Naive FLAIM, this is likely worsened by client heterogeneity. Further, we observe that for AugFLAIM (Private), the utility matches that of DistAIM unless the choice of $T$ is very large. At $T = 100$, the variance in utility is high, sometimes even worse than that of NaiveFLAIM. This is since the privacy cost scales in both the number of rounds and features, resulting in too much noise. In the case of annealing, $T$ is chosen adaptively by an early stopping condition (see Appendix B.4). Compared to AIM, budget annealing obtains sub-optimal utility across all federated methods. For annealing on Adult, AugFLAIM (Private) matches AugFLAIM (Non-private) and both perform better than NaiveFLAIM. Overall, we found choosing $T$ to be small gives best performance for AugFLAIM over a our datasets and should avoid using budget annealing. We explore this further in Appendix C.

**Client-participation** $(p)$. In Figure 2c, we plot the average workload error whilst varying the per-round client participation rate $(p)$ with $T = 10, \varepsilon = 1$. We observe clearly the gap in performance between central AIM and DistAIM is caused by the error introduced by subsampling and when $p = 0.5$ performance is almost matched. For NaiveFLAIM, we observe the performance improvement as $p$ increases is much slower than other methods. Since when $p$ is large, NaiveFLAIM receives many measurements, each of which are likely to be highly heterogeneous and thus the model struggles to learn consistently. For both AugFLAIM variations, we observe the performance increases with client participation but does eventually plateau. AugFLAIM (Private) consistently matches the error of DistAIM except when $p$ is large, but we note this is not a practical regime in FL.

Table 1: Performance on datasets $K = 100, p = 0.1, \varepsilon = 1, T = 10$. Results show workload error and negative log-likelihood. Metrics are in bold if an FL method achieves lowest on a specific dataset.

| Method / Dataset | Adult | Magic | Mushroom | Nursery | SynthFS |
|---|---|---|---|---|---|
| AugFLAIM | 0.8 / 29.28 | 1.64 / 2587.5 | 1.48 / 8562.27 | 1.33 / 6032.65 | 0.94 / 26.21 |
| AugFLAIM (Non-private) | 0.62 / 23.91 | 1.18 / 62.84 | 0.92 / 39.21 | 0.83 / 118.41 | 0.3 / 16.92 |
| AugFLAIM (Private) | 0.43 / 21.74 | **1.07 / 28.9** | 0.79 / **23.9** | 0.46 / **11.94** | 0.26 / 16.92 |
| DistAIM | **0.42 / 21.41** | 1.08 / 35.04 | **0.65** / 37.14 | **0.36** / 15.47 | **0.25 / 16.91** |
| AIM | 0.2 / 19.3 | 0.85 / 23.7 | 0.38 / 16.87 | 0.05 / 9.7 | 0.09 / 15.73 |

**Varying heterogeneity** ($\beta$). In Figure 2d, we plot the average workload error on the Adult dataset over client splits formed by varying the heterogeneity parameter ($\beta$) to produce label-skew. Here, a larger $\beta$ corresponds to a more uniform partition and therefore less heterogeneity. In the label-skew setting, data is both skewed according to the class attribute of Adult and the number of samples, with only a few clients holding the majority of the dataset. We observe that when the skew is large ($\beta < 0.1$), all methods struggle. As $\beta$ increases and skew decreases, NaiveFLAIM performs the worst and AugFLAIM (Private) has stable error, close to that of DistAIM.

**Varying local rounds** ($s$). A benefit of the federated setting is that clients can perform a number of local steps before sending all measured marginals to the server. However, for FLAIM methods, this incurs an extra privacy cost in the number of local rounds ($s$). In Figure 2e, we vary $s \in \{1, 4\}$ and plot the workload error. We observe that although there is an associated privacy cost with increasing $s$, the

Table 2: Overhead of DistAIM vs. FLAIM at optimal $T$

| | $T(\uparrow)$ | Bandwidth ($\uparrow$) | Err ($\downarrow$) | NLL ($\downarrow$) |
|---|---|---|---|---|
| Adult | 2× | 1300× | 0.58× | 0.1× |
| Magic | 3.2× | 1643× | 0.19× | 0.15× |
| Mushroom | 7× | 7.5× | 0.79x | 0.4× |
| Nursery | 20× | 3.4× | 0.89× | 0.17× |

workload errors are not significantly different for small $T$. As we vary $T$, the associated privacy cost becomes larger and the workload error increases for methods that are performing $s = 4$ local updates. Although increasing $s$ does not result in lower workload error, and in cases where $T$ is misspecified can give far worse performance, it is instructive to instead study the test AUC of a classification model trained on the synthetic data. In Figure 2f we see that performing more local updates can give better test AUC after fewer global rounds. For AugFLAIM (Private), this allows us to match the test AUC performance of DistAIM. We note similar behaviour for AugFLAIM (Non-private).

**Comparison across datasets.** Table 1 presents results across our benchmark datasets with client data partitioned via the clustering approach (except SynthFS which is constructed with feature-skew). We set $\varepsilon = 1, p = 0.1$ and $T = 10$. For each method we present both the average workload error and the negative log-likelihood over a holdout set. The first is a form of training error and the second a measure of generalisation. We observe that on 3 of the 5 datasets AugFLAIM (Private) achieves the lowest negative log-likelihood and for workload error, closely matches DistAIM on 3 datasets.

**Distributed vs. Federated AIM.** Table 2 presents the overhead of DistAIM when compared to AugFLAIM (Private) including the average client bandwidth (sent and received communication) across protocols. We set $T \in [1, 200]$ that achieves lowest workload error. Observe on Adult, DistAIM requires twice as many rounds to achieve optimal error and results in a large (1300×) increase in client bandwidth compared to AugFLAIM. However, this results in 2× lower workload error and a 10% improvement in NLL. This highlights one of the chief advantages of FLAIM, which for a small loss in utility, we can obtain much lower overheads. Furthermore, while a 2× gap in workload error seems significant, we refer back to Figure 2f, which shows the resulting classifier has AUC which is practical for downstream tasks. We note the client bandwidth is only significantly larger when the total workload dimension is large (e.g., on Adult and Magic). For datasets with much smaller feature cardinalities, there is still communication overhead but it is not as significant.

**Conclusion.** We have shown that naively federating AIM under the challenges of FL causes a large decrease in utility when compared to the SMC-based DistAIM. To counteract this, we propose AugFLAIM (Private), which augments local decisions with a proxy for heterogeneity and obtains utility close to DistAIM while lowering overheads. In the future, we plan to extend our approaches to support user-level DP where clients hold multiple data items related to the same individual.

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

---

**Algorithm 2** AIM (McKenna et al., 2022)

---

**Input:** Dataset $D \in \mathbb{R}^{N \times d}$, workload $Q$, privacy parameters $\varepsilon, \delta$, Maximum model size $S$
**Output:** Synthetic dataset $\hat{D}$
 1: Initialise the zCDP budget $\rho_{\text{total}} \leftarrow \rho(\varepsilon, \delta)$ via Lemma A.2
 2: Set $\sigma_0^2 \leftarrow \frac{16d}{0.9 \cdot \rho_{\text{total}}}, \varepsilon_0 \leftarrow \sqrt{8 \cdot 0.1 \cdot \rho_{\text{total}}/16d}$
 3: Set $t \leftarrow 0$ and $Q \leftarrow \text{Completion}(Q)$
 4: For each marginal $q \in Q$ initialise weights $w_q := \sum_{r \in Q} |q \cap r|$
 5: **while** $\rho_{\text{used}} < \rho_{\text{total}}$ **do**
 6:     $t \leftarrow t + 1$
 7:     **if** $t = 0$ **then**                                                     ▷ Initialise with one-way marginals
 8:         Filter current rounds workload as
$$Q_0 \leftarrow \{q \in Q : |q| = 1\}$$
 9:         Measure $\tilde{M}_q(D) \leftarrow M_q(D) + N(0, \sigma_0^2 I), \forall q \in Q_0$ and use PGM to estimate $\hat{D}_0$
10:         $\rho_{\text{used}} \leftarrow \rho_{\text{used}} + \frac{d}{2\sigma_t^2}$
11:     **else**
12:         Filter the workload
$$Q_t \leftarrow \{q \in Q : \text{ModelSize}(\hat{D}_{t-1}, q) \leq \frac{\rho_{\text{used}}}{\rho_{\text{total}}} \cdot S\}$$
13:         **Select** $q_t \in Q_t$ using the exponential mechanism with parameter $\varepsilon_t$ and utility function
$$u(q; D) \leftarrow w_q \cdot \left( \|M_q(D) - M_q(\hat{D}_{t-1})\|_1 - \sqrt{\frac{2}{\pi}} \cdot \sigma_t \cdot n_q \right)$$
14:         **Measure** the chosen marginal $q_t$ with the Gaussian mechanism i.e.,
$$\tilde{M}_{q_t}(D) \leftarrow M_{q_t}(D) + \mathcal{N}(0, \sigma_t^2 I)$$
15:         **Estimate** the new model via PGM (McKenna et al., 2019)
$$\hat{D}_t \leftarrow \arg\min_{p \in \mathcal{S}} \sum_{i=1}^{t} \frac{1}{\sigma_i} \|M_{q_i}(p) - \tilde{M}_{q_i}(D)\|_2$$
16:         $\rho_{\text{used}} \leftarrow \rho_{\text{used}} + \frac{\varepsilon_t^2}{8} + \frac{1}{2\sigma_t^2}$
17:     **if** $\|M_{q_t}(\hat{D}_t) - M_{q_t}(\hat{D}_{t-1})\|_1 \leq \sqrt{2/\pi} \cdot \sigma_t \cdot n_{q_t}$ **then**                    ▷ Budget Annealing
18:         Set $\sigma_{t+1} \leftarrow \sigma_t/2, \varepsilon_{t+1} \leftarrow 2 \cdot \varepsilon_t$
19:     **if** $(\rho_{\text{total}} - \rho_{\text{used}}) \leq 2(1/2\sigma_{t+1}^2 + \frac{1}{8}\varepsilon_{t+1}^2)$ **then**                              ▷ Final round
20:         Set $\sigma_{t+1}^2 \leftarrow 1/(2 \cdot 0.9 \cdot (\rho_{\text{total}} - \rho_{\text{used}})), \varepsilon_{t+1} \leftarrow \sqrt{8 \cdot 0.1 \cdot (\rho_{\text{total}} - \rho_{\text{used}})}$
21: **return** $\hat{D}_t$

---

# A    ALGORITHM DETAILS

## A.1    AIM

The current SOTA method, and the core of our federated algorithms is AIM, introduced by McKenna et al. (2022). AIM extends the main ideas of MWEM (Hardt et al., 2012) but augments the algorithm with an improved utility score function, a graphical model-based inference approach via PGM and more efficient privacy accounting with zero-Concentrated Differential Privacy (zCDP). The full details of AIM are outlined in Algorithm 2. We refer to this algorithm as 'Central AIM', to distinguish it from the distributed and federated versions we consider in the main body of the paper. It is important to highlight the following details:

- **zCDP Budget Initialisation:** In central AIM, the number of global rounds $T$ is set adaptively via budget annealing. To begin, $T := 16 \cdot d$ where $d$ is the number of features. This is the

maximum number of rounds that will occur in the case where the annealing condition is never triggered. This initialisation occurs in Line 2.

- **Workload Filtering:** The provided workload of queries, $Q$, is extended by forming the completion of $Q$. That is to say, all lower order marginals contained within any $q \in Q$ are also added to the workload. Furthermore, for the first round the workload is filtered to contain only 1-way marginals to initialise the model. This occurs in Line 8. In subsequent rounds, the workload is filtered to remove any queries that would force the model to grow beyond a predetermined maximum size $S$. This occurs at Line 12.

- **Weighted Workload:** Each marginal $q \in Q$ is assigned a weight via $w_q = \sum_{r \in Q} |q \cap r|$. Thus, marginals that have high overlap with other queries in the workload are more likely to be chosen. This is computed in Line 4.

- **Model Initialisation**: Instead of initialising the synthetic distribution uniformly over the dataset domain, the synthetic model is initialised by measuring each 1-way marginal in the workload $W$ and using PGM to estimate the initial model. This corresponds to measuring each feature's distribution once before AIM begins and occurs in Lines 7-10.

- **Query Selection:** A marginal query is selected via the exponential mechanism with utility scores that compare the trade-off between the current error and the expected error when measured under Gaussian noise. The utility scores and selection step occur at Line 13.

- **Query Measurement:** Once a query has been chosen, it is measured under the Gaussian mechanism. This occurs at Line 14.

- **PGM model estimation:** The current PGM model is updated by adding the newly measured query to the set of previous measurements. The PGM model parameters are then updated by a form of mirror descent for a number of iterations. The precise details of PGM can be found in McKenna et al. (2019). This occurs at Line 15.

- **Budget Annealing:** At the end of every round, the difference between the measured query of the new model and that of the previous model is taken. If this change is smaller than the expected error under Gaussian noise, the noise parameters are annealed by halving the amount of noise. This occurs at Line 17. If after this annealing there is only a small amount of remaining privacy budget left, the noise parameters can instead be calibrated to perform one final round before finishing. This occurs at Line 20.

## A.2 DISTAIM

We describe in full detail the DistAIM algorithm introduced in Section 3 and outlined in Algorithm 3. The algorithm can be seen as a straightforward extension of Pereira et al. (2022) who propose a secure multi-party computation (SMC) approach for distributing MWEM. The key differences are that we replace MWEM with AIM and consider a federated setting where not all participants are available at any particular round. The approach relies on participants secret-sharing their query answers to compute servers who then perform a number of SMC operations over these shares to train the model. The resulting algorithm is identical to AIM in outline but has a few subtle differences:

- **Secret Sharing:** Participants must secret-share the required quantities to train AIM. In Pereira et al. (2022), it is assumed that the full workload answers $\{\llbracket M_q(D) \rrbracket : q \in Q\}$ have already been secret-shared between a number of compute servers. In DistAIM, we assume that clients sampled to participate at a particular round contribute their secret-shared workload answers $\{\llbracket M_q(D_k) \rrbracket : q \in Q\}$ which are aggregated with the shares of current and past participants from previous rounds. Thus, as the number of global rounds $T$ increases, the secret-shared answers approach that of the central dataset. We assume the same SMC framework as Pereira et al. (2022) which is a 3-party scheme based on Araki et al. (2016).

- **Client participation:** At each round only a subset of the participants are available to join the AIM round. In expectation $pK$ clients will contribute their local marginals $\llbracket M_q(D_k) \rrbracket$ in the form of secret-shares. Compared to the central setting, DistAIM incurs additional error due to this subsampling.

- **Select step:** One key obstacle in extending AIM to a distributed setting is the exponential mechanism. Since each client holds a local dataset $D_k$, they cannot share their data with

---

**Algorithm 3** DistAIM

---

**Input:** Participants $P_1, \ldots P_k$ with local datasets $D_1, \ldots, D_k$, privacy parameters $(\varepsilon, \delta)$
1: Initialise AIM parameters as in Lines 1-4 of Algorithm 2
2: **for** each round $t$ **do**
3:     Sample participants $P_t \subseteq [k]$ with probability $p$ and remove those who have already participated
4:     For each $k \in P_t$ who have not participated before, secret-share the workload answers $\{[\![M_q(D_k)]\!] : q \in W\}$ to the compute servers (Araki et al., 2016)
5:     **Aggregate:** The compute servers aggregate shares of the received answers and combine with previously received shares $[\![M_q(\tilde{D}_t)]\!] := \sum_{i=1}^{t-1} \sum_{k \in P_i} [\![M_q(D_k)]\!] + \sum_{k \in P_t} [\![M_q(D_k)]\!]$
6:     **Select:** Compute servers select $q_t \in Q$ using the exponential mechanism over secret shares $[\![M_q(\tilde{D}_t)]\!]$ via Protocol 2 in Pereira et al. (2022) with AIM utility scores

$$u(q; D_t) := w_q \cdot (\|M_q(\tilde{D}_t) - M_q(\hat{D}_{t-1})\|_1 - \sqrt{\frac{2}{\pi}} \cdot \sigma_t \cdot n_q)$$

7:     **Measure:** $q_t$ is measured using $[\![M_{q_t}(\tilde{D}_t)]\!]$ under a variation of Protocol 3 in Pereira et al. (2022), replacing Laplace noise with Gaussian to produce $\tilde{M}_{q_t}(\tilde{D}_t)$
8:     **Estimate** the new model via PGM using the received noisy measurements e.g.

$$\hat{D}_t \leftarrow \arg\min_{p \in \mathcal{S}} \sum_{i=1}^{t} \frac{1}{\sigma_i} \|M_{q_i}(p) - \tilde{M}_{q_i}(\tilde{D}_i)\|_2$$

---

    the central server. Instead the quality functions $u(q; D)$ must be computed in a distributed manner between the compute servers who hold shares of the workload answers.

- **Measure step:** Once the marginal $q_t$ has been selected by a secure exponential mechanism, it must be measured. As Pereira et al. (2022) utilise MWEM, they measure queries under Laplace noise which can be easily generated in an SMC setting. AIM instead uses Gaussian noise and this is also what we use in DistAIM. In practice, one can implement this easily under SMC e.g., using the Box-Muller method.

## A.3   FLAIM: PRIVACY GUARANTEES

In this section, we present and prove the privacy guarantees of the FLAIM approach. For completeness, we provide additional definitions and results, starting with the definition of $(\varepsilon, \delta)$-Differential Privacy.

**Definition A.1** (Differential Privacy (Dwork and Roth, 2014)). *A randomised algorithm $\mathcal{M} \colon \mathcal{D} \to \mathcal{R}$ satisfies $(\varepsilon, \delta)$-differential privacy if for any two adjacent datasets $D, D' \in \mathcal{D}$ and any subset of outputs $S \subseteq \mathcal{R}$,*

$$\mathbb{P}(\mathcal{M}(D) \in S) \leq e^\varepsilon \mathbb{P}(\mathcal{M}(D') \in S) + \delta.$$

While we work using the more convenient formulation of $\rho$-zCDP (Definition 2.3), it is common to translate this guarantee to the more interpretable $(\varepsilon, \delta)$-DP setting via the following lemma.

**Lemma A.2** (zCDP to DP (Canonne et al., 2020)). *If a mechanism $\mathcal{M}$ satisfies $\rho$-zCDP then it satisfies $(\varepsilon, \delta)$-DP for all $\varepsilon > 0$ with*

$$\delta = \min_{\alpha > 1} \frac{\exp((\alpha - 1)(\alpha\rho - \varepsilon))}{\alpha - 1} \left(1 - \frac{1}{\alpha}\right)^\alpha$$

We now restate the privacy guarantees of FLAIM and its variations.

**Lemma A.3** (Lemma 4.1 restated). *For any number of global rounds $T$ and local rounds $s$, FLAIM satisfies $(\varepsilon, \delta)$-DP , under Gaussian budget allocation $r \in (0, 1)$ by computing $\rho$ according to Lemma A.2, and setting*

$$\sigma_t = \begin{cases} \sqrt{\frac{Ts+d}{2 \cdot r \cdot \rho}}, & \textit{Naive or Non-private} \\ \sqrt{\frac{T(s+d)}{2 \cdot r \cdot \rho}}, & \textit{Private} \end{cases} \quad , \quad \varepsilon_t = \sqrt{\frac{8 \cdot (1 - r) \cdot \rho}{Ts}}$$

Table 3: Datasets

| Dataset | # of training samples | # of features | Class Imbalance |
|---|---|---|---|
| Adult (Kohavi and Becker, 1996) | 43,598 | 14 | 0.27 |
| Magic (Bock, 2007) | 17,118 | 11 | 0.35 |
| Mushroom (Schlimmer, 1987) | 7312 | 22 | 0.49 |
| Nursery (Rajkovic, 1997) | 11,663 | 9 | 0.31 |
| SynthFS (see Appendix B.1.1) | 45,000 | 10 | N/A |

*For AugFLAIM methods, the exponential mechanism is applied with sensitivity $\Delta := \max_q 2w_q$.*

*Proof.* For NaiveAIM, the result follows almost directly from AIM, since $T$ rounds in the latter correspond to $T \cdot s$ in the former. We then apply the existing privacy bounds for AIM. Similarly, for AugFLAIM (Private), the 1-way marginals of every feature are included in the computation, thus increasing the number of measured marginals under Gaussian noise to $T \cdot (s + d)$. In all variations, the exponential mechanism is only applied once for each local round and thus $Ts$ times in total. For AugFLAIM, the augmented utility scores $u(q; D_k)$ lead to a doubling of the sensitivity compared to AIM, since $M_q(D_k)$ is used twice in the utility score and thus $\Delta := 2 \cdot \max_q w_q$. □

## B  EXPERIMENTAL SETUP

### B.1  DATASETS

In our experiments we use a range of tabular datasets from the UCI repository (Dua and Graff, 2017) and one synthetic dataset that we construct ourselves. A summary of all datasets in terms of the number of training samples, features and class imbalance is detailed in Table 3. All datasets are split into a train and test set with $90\%$ forming the train set. From this, we form clients local datasets via a partitioning method (see Appendix B.2). In more detail:

- **Adult** — A census dataset that contains information about adults and their occupations. The goal of the dataset is to predict the binary feature of whether their income is greater than $50,000. The training set we use contains 43,598 training samples and 14 features.

- **Magic** — A dataset on imaging measurements from a telescope. The classification task is to predict whether or not the measurements are signal or background noise. The training set we use contains 17,118 samples and 11 features.

- **Mushroom** — This dataset contains hypothetical measurements of mushrooms. The goal is to predict whether the mushroom is edible or poisonous. The training set we use contains 7312 samples and 22 features.

- **Nursery** — A dataset formed from nursery applications. The classification task is to predict whether or not to admit a child to the nursery. The training set we use contains 11,663 samples and 9 features.

- **SynthFS** — A synthetic dataset formed from sampling features from a Gaussian distribution with different means. The precise construction is detailed in Appendix B.1.1. In our experiments, the training set contains 45,000 samples with 10 features.

All continuous features are binned uniformly between the minimum and maximum which we assume to be public knowledge. We discretize our features with 32 bins, although experiments varying this size presented no significant change in utility. This follows the pre-processing steps taken by prior work (McKenna et al., 2022; Aydore et al., 2021).

### B.1.1  SYNTHFS

In order to simulate feature-skew in an ideal setting for FLAIM, we construct a synthetic dataset that we denote SynthFS. To create SynthFS, we draw independent features from a Gaussian distribution

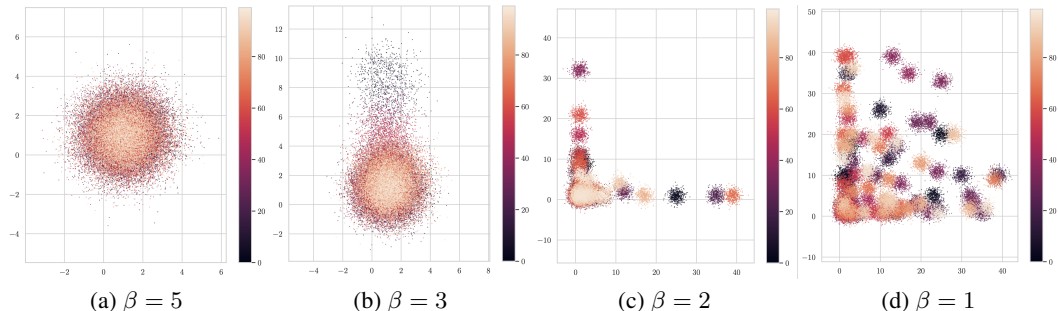

(a) $\beta = 5$      (b) $\beta = 3$      (c) $\beta = 2$      (d) $\beta = 1$

Figure 3: SynthFS: Synthetic dataset constructed with feature skew, varying $\beta \in \{1, 2, 3, 5\}$

where the mean is chosen randomly from a Zipfian distribution whose parameter $\beta$ controls the skew. This is done in the following manner:

- For each client $k \in [K]$ and feature $m \in [d]$ sample mean $\mu_m^k \sim \mathrm{Zipf}(\beta, n_{\mathrm{zipf}})$
- For each feature $m \in [d]$, sample $n/K$ examples for client $k$ from $N(\mu_m^k, 1)$

In our experiments we set $n = 50,000$ such that for $K = 100$ each client is assigned 500 samples. In order to form a test set we sample 10% from the dataset and assign the rest to clients. We fix $d = 10$ and $n_{\mathrm{zipf}} = 40$ in all constructions. We highlight this process for $\beta \in \{1, 2, 3, 5\}$ in Figure 3, with $d = 2$ features for visualization purposes only. By increasing $\beta$, we decrease the skew of the means being sampled from the Zipf distribution. Hence, for larger $\beta$ values, each clients features are likely to be drawn from the same Gaussian and there is no heterogeneity. Decreasing $\beta$ increases the skew of client means and each feature is likely to be drawn from very different Gaussian distributions as shown when $\beta = 1$.

## B.2 HETEROGENEITY: NON-IID CLIENT PARTITIONS

In order to simulate heterogeneity on our benchmark datasets, we take one of the tabular datasets outlined in Appendix B.1 and form partitions for each client. The aim is to create client datasets that exhibit strong data heterogeneity by varying the number of samples and inducing feature-skew. We do this in two ways:

- **Clustering Approach** — In the majority of our experiments, we form client partitions via dimensionality reduction using UMAP (McInnes et al., 2018). An example of this process is shown in Figure 4 for the Adult dataset. Figure 4a shows a UMAP embedding of the training dataset in two-dimensions where each client partition (cluster) is highlighted a different color. To form these clusters we simply use $K$-means where $K = 100$ is the total number of clients we require. In Figures 4b-4d, we display the same embedding but colored based on different feature values for age, hours worked per-week and income $> 50$k. We observe, for instance, the examples that are largest in age are concentrated around $x = 10$ while those who work more hours are concentrated around $y = -7$. Thus clients that have datasets formed from clusters in the area of $(10, -7)$ will have significant feature-skew with a bias towards older adults who work more hours. These features have been picked at random and other features in the dataset have similar skew properties. The embedding is used only to map the original data to clients, and the raw data is used when training AIM models.

- **Label-skew Approach** — While the clustering approach works well to form non-IID client partitions, there is no simple parameter to vary the heterogeneity of the partitions. In experiments where we wish to vary heterogeneity, we follow the approach outlined by Li et al. (2022). For each value the class variable can take, we sample the distribution $p_C \sim \mathrm{Dirichlet}(\beta) \subseteq [0, 1]^K$ and assign examples with class value $C$ to the clients using this distribution. This produces client partitions that are skewed via the class variable of the dataset, where a larger $\beta$ decreases the skew and thus reduces heterogeneity.

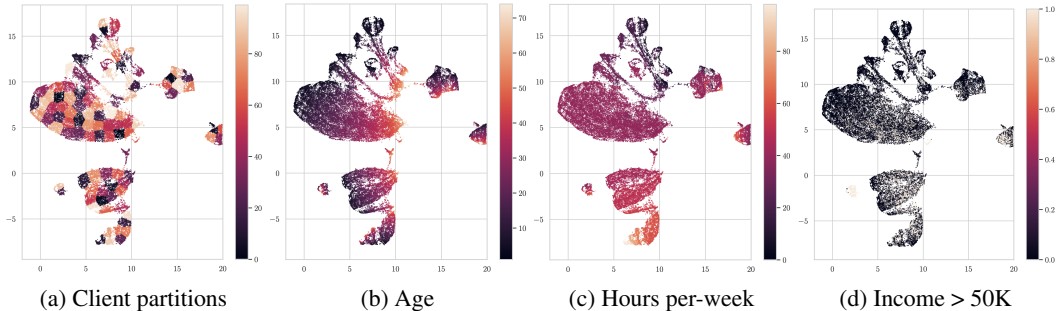

| (a) Client partitions | (b) Age | (c) Hours per-week | (d) Income > 50K |

Figure 4: Clustering approach to form non-IID splits on Adult dataset, $K = 100$ clients. All plots show the same embedding formed from UMAP, with Figure 4 highlighting each client's local dataset which has been formed by clustering in the embedding space. Figures 4b-4d show the same embedding but colored based on three features: age, hours worked per-week and income > 50k. The embedding is used only to map examples to clients, and AIM models are trained on the raw data.

Table 4: Average heterogeneity $\frac{1}{K} \sum_k \sum_{q \in Q} \tau_k(q)$ across client partition methods with $K = 100$.

| Dataset / Partition | IID | Clustering | Label-skew ($\beta = 0.1$) | Label-skew ($\beta = 0.8$) |
|---|---|---|---|---|
| Adult | 0.241 | 0.525 | 0.531 | 0.332 |
| Magic | 0.538 | 0.792 | 0.767 | 0.603 |
| Mushroom | 0.181 | 0.662 | 0.559 | 0.301 |
| Nursery | 0.175 | 0.701 | 0.530 | 0.289 |

Table 4 presents the average heterogeneity for a fixed workload of queries across each of the datasets and the different partition methods with $K = 100$ clients. We look at the following methods: IID sampling, clustering approach, label-skew with $\beta = 0.1$ (large-skew) and label-skew with $\beta = 0.8$ (small-skew). Observe in all cases, our non-IID methods have higher heterogeneity than IID sampling. Specifically, the clustering approach does well to induce heterogeneity and often results in twice as much query skew across the workload. Note also that increasing $\beta$ from 0.1 to 0.8 decreases average heterogeneity and in some cases at $\beta = 0.8$, the heterogeneity is close to IID sampling. This confirms that simulating client partitions in this way is useful for experiments where we wish to vary heterogeneity, since we can vary $\beta$ accordingly and $\beta \in (0, 1]$ in experiments is well-chosen.

### B.3 EVALUATION

In our experiments we evaluate our methods with three different metrics:

**1. Average Workload Error.** We mainly evaluate (FL)AIM methods via the average workload error. For a fixed workload of marginal queries $Q$, we measure $\text{Err}(D, \hat{D}; Q) := \frac{1}{|Q|} \sum_{q \in Q} \|M_q(D) - M_q(\hat{D})\|_1$ where $D := \cup_k D_k$. This can be seen as a type of training error since the models are trained to answer the queries in $Q$.

**2. Negative Log-likelihood.** An alternative is the (mean) negative log-likelihood of the synthetic dataset sampled from our (FL)AIM models when compared to a heldout test set. This metric can be viewed as a measure of generalisation, since the metric is agnostic to the specific workload chosen.

**3. Test ROC-AUC.** In specific cases we evaluate our AIM models by training a gradient boosted decision tree (GBDT) model on the synthetic data it produces. All our datasets are binary classification tasks. We test the performance of this model on a holdout set and evaluate the ROC-AUC.

### B.4 ADDITIONAL HYPERPARAMETERS FOR (FL)AIM

**PGM Iterations:** The number of PGM iterations determines how many optimisation steps are performed to update the parameters of the graphical model during training. AIM has two parameters, one for the number of training iterations between global rounds of AIM and one for the final number

of iterations performed at the end of training. We set this to 100 training iterations and 1000 final iterations. This is notably smaller than the default parameters used in central AIM, but we have verified that there is no significant impact on performance in utility.

**Model Initialisation:** We follow the same procedure as in central AIM, where every 1-way marginal is estimated to initialise the model. Instead in our federated settings, we take a random sample of clients and have them estimate the 1-way marginals and initialise the model from these measurements.

**Budget Annealing Initialisation:** When using budget annealing, the initial noise is calibrated under a high number of global rounds. In central AIM, initially $T = 16 \cdot d$ resulting in a large amount of noise until the budget is annealed. We instead set this as $T = 8 \cdot d$ since empirically we have verified that a smaller number of global AIM rounds is better for performance in the federated setting.

**Budget Annealing Condition:** In central AIM, the budget annealing condition compares the previous model estimate with the new model estimate of the current marginal. If the annealing condition is met, the noise parameters are decreased. In the federated setting, it is possible that PGM receives multiple new marginals at a particular round. We employ the same annealing condition, except we anneal the budget if at least one of the marginals received from the last round passes the check.

## C  FURTHER EXPERIMENTS

**Varying $\varepsilon$:** In Figures 7a – 7c, we vary $\varepsilon$ across the Magic, Mushroom and Nursery datasets under a clustering partition. This plot replicates Figure 2a across three other datasets. We observe fairly similar patterns to that of Figure 2a with NaiveFLAIM performing the worst across all settings, and our AugFLAIM methods helping to correct for this and closely match the performance of DistAIM. There are however some consistent differences when compared to the Adult datasets. For example, on the Magic and Mushroom dataset, AugFLAIM (Private) performance comes very close to DistAIM but there is a consistent gap in workload error. This is in contrast to the Adult dataset where AugFLAIM (Private) shows a more marked improvement over DistAIM. For Nursery, the results are similar to Adult with AugFLAIM matching the error of DistAIM.

**Varying $T$:** In Figures 7d – 7f, we vary the number of global rounds $T$ while fixing $\varepsilon = 1$ and $K = 100$ clients under a clustering partition. This replicates Figure 2b but over the other datasets. Across all figures we plot dashed lines to show the mean workload error under the setting where $T$ is chosen adaptively via budget annealing. On datasets other than Adult, we observe more clearly that the choice of $T$ is very significant to the performance of AugFLAIM (Private) and choosing $T > 30$ results in a large increase in workload error across all datasets. In contrast, increasing $T$ for DistAIM often gives an improvement to the workload error. Recall, that DistAIM has participants secret-share their workload answers and these are aggregated over a number of rounds. Hence, as $T$ increases the workload answers DistAIM receives approaches that of the central dataset. As we can see on Nursery, at $T = 100$ the utility of DistAIM almost matches that of central AIM. For budget annealing, on two of the three datasets, AugFLAIM (Private) has improved error over NaiveFLAIM but does not always result in performance that matches DistAIM. Instead, it is recommended to choose a much smaller $T \in [5, 30]$ which has consistently good performance across all four of the datasets.

**Varying $p$:** In Figures 7g – 7i, we vary the participation rate $p$ while fixing $\varepsilon = 1, T = 10$ and $K = 100$ clients under a clustering partition. This replicates Figure 2c but across the other datasets. We observe similar patterns as we did on Adult. DistAIM approaches the utility of central AIM as $p$ increases. The workload error on the Magic and Mushroom datasets stabilises for AugFLAIM methods when $p > 0.3$. Generally, when $p$ is large, DistAIM is preferable but we note this doesn't correspond to a practical federated setting where sampling rates are typically much smaller ($p < 0.1$).

**Varying $\beta$** In Figures 7j – 7l, we vary the label-skew partition across the three datasets via the parameter $\beta$. These experiments replicate that of Figure 2d. As before, we clearly observe that NaiveFLAIM is subject to poor performance and that this is particularly the case when there is high skew (small $\beta$) in participants datasets. We can see that the AugFLAIM methods help to stabilise performance and when skew is large ($\beta < 0.1$) can help match DistAIM across the three datasets. We note that on Nursery there is a consistent utility gap between the AugFLAIM methods and DistAIM.

**Local updates:** In Figure 8 we vary the local updates $s \in \{1, 4\}$ while fixing $\varepsilon = 1, T = 10$ and $K = 100$. This replicates Figure 2e and 2f but across the other datasets. When using $s = 4$ local

Table 5: **Budget annealing ranking** across workload error and negative log-likelihood. Ranks are averaged across four datasets, with each method repeated 10 times. $T$ is set adaptively via annealing.

| Method / $\varepsilon =$ | 1 | 2 | 3 | 5 |
|---|---|---|---|---|
| NaiveFLAIM | 4.65 / 4.75 | 4.875 / 4.9 | 4.975 / 4.95 | 5.0 / 5.0 |
| AugFLAIM (Non-private) | 3.6 / 3.3 | 3.85 / 3.525 | 3.9 / 3.775 | 3.925 / 3.6 |
| AugFLAIM (Private) | 3.75 / 3.45 | 3.25 / 3.15 | 3.125 / 3.125 | 3.05 / 3.25 |
| DistAIM | 2.0 / 2.5 | 2.025 / 2.425 | 2.0 / 2.15 | 2.025 / 2.15 |
| AIM | 1.0 / 1.0 | 1.0 / 1.0 | 1.0 / 1.0 | 1.0 / 1.0 |

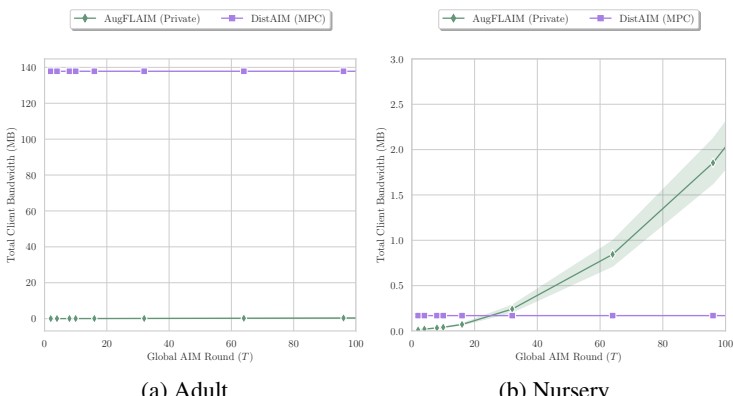

(a) Adult                    (b) Nursery

Figure 5: Total client bandwidth across DistAIM and AugFLAIM (Private) when varying $T$

rounds the workload error across methods often increases for NaiveFLAIM and AugFLAIM methods. However, when looking at the test AUC performance, taking $s = 4$ local updates often gives better AUC performance than $s = 1$ on the Magic and Nursery datasets. This results in AUC that is closer to that of DistAIM than the other FLAIM methods but there is still a notable gap.

**Budget Annealing:** In Table 5 we present the average rank of methods across four datasets: Adult, Magic, Mushroom and Nursery, all partitioned via clustering. We rank based on two metrics: workload error and negative log-likelihood. The number of rounds $T$ is set adaptively via budget annealing. We vary $\varepsilon \in \{1, 2, 3, 4, 5\}$ with the goal of understanding how annealing affects utility across methods. First note that DistAIM achieves the best rank across all settings when using budget annealing, only beaten by central AIM. When $\varepsilon$ is small, AugFLAIM (Non-private) achieves a better average ranking across both metrics when compared to AugFLAIM (Private). However, as $\varepsilon$ increases, AugFLAIM (Private) achieves better rank, only beaten by DistAIM. AugFLAIM (Private) can achieve better performance by choosing $T$ to be reasonably small ($T < 30$) as previously mentioned.

**DistAIM vs. FLAIM Communication:** In Figure 5, we present the average client bandwidth (total sent and received) on Adult and Nursery. In DistAIM, the amount of communication a client does is constant no matter the value of $T$, since they only send secret-shared answers once. We observe two clear scenarios: In the case where the total dimension of a workload is large, the gap in client bandwidth between AugFLAIM and DistAIM is also large. On Adult, clients must send $\sim 140$Mb in shares whereas AugFLAIM is an order of magnitude smaller at a few megabytes. Sending 140Mb of shares may not seem prohibitive but this size quickly scales in the cardinality of features and in practice could be large e.g., on datasets with many continuous features discretized to a reasonable number of bins. For Nursery, the cardinality of features is small. Hence, communication overheads in DistAIM are very small ($< 1$Mb). AugFLAIM overtakes DistAIM in communication when $T > 30$. Note however that setting $T > 30$ in AugFLAIM gives far worse utility than DistAIM, so exceeding DistAIM bandwidth would not occur in practice since one would choose $T < 30$ (see Figure 7f).

**Ablation on heterogeneity measures:** In Section 5, we show AugFLAIM (Private) has best performance when compared with all FLAIM variants including that of AugFLAIM (Non-private). In order to understand where the performance improvements of AugFLAIM (Private) come from we

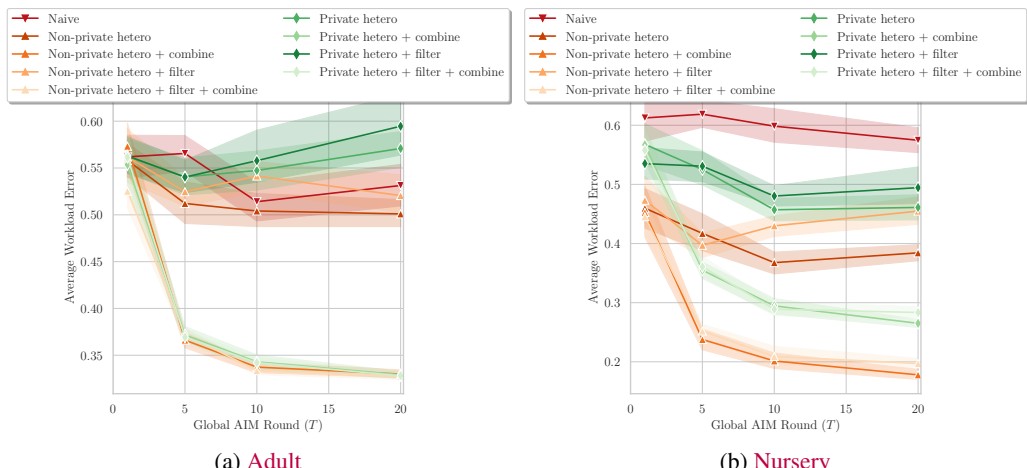

(a) Adult                (b) Nursery

Figure 6: Ablation study - We plot the various modifications to NaiveFLAIM that make up our AugFLAIM methods. We plot the L1 workload error for $\varepsilon = 5$ on Adult and Nursery.

perform an ablation by breaking the methods down into various modifications over NaiveFLAIM which include:

- **+ Non-private Hetero** - Use the true heterogeneity measurement $\tau_k(q)$ introduced in Section 4.2 when computing utility scores $u(q; D)$.

- **+ Private Hetero** - Use the private heterogeneity measurement $\tilde{\tau}_k(q)$ introduced in Section 4.3 when computing utility scores $u(q; D)$.

- **+ Combine** - Privately estimate all 1-way marginals at every global round and feed these back to the PGM model. For (Private hetero + combine) this is obtained without additional privacy cost. When used with (Non-private hetero) additional privacy budget is spent on estimating 1-ways at each round but the true heterogeneity $\tau_k(q)$ is used in utility scores.

- **+ Filter** - Remove 1-way marginals from the workload. For (... + Combine + Filter) this is done to prevent clients from measuring 1-way marginals multiple times in a single global round.

In our experiments, AugFLAIM (Private) is equivalent to (Private Hetero + Combine + Filter) whereas AugFLAIM (Non-private) is equivalent to (Non-private Hetero) with no combining or filtering. One of the key reasons for the utility improvement in AugFLAIM (Private) over AugFLAIM (Non-private) in our experiments is that 1-way marginals are repeatedly estimated during training. This seems to lower the overall workload error, possibly forcing the model to produce answers to higher order marginals that are more consistent with the 1-ways.

In Figure 6, we perform an ablation by taking NaiveFLAIM and adding combinations of the modifications outlined above. We study the workload error on the Adult and Nursery datasets. For Nursery, we observe when we add the (Filter + Combine) modifications to the non-private heterogeneity measure $\tau_k(q)$, it achieves best performance over all methods including lower error than the fully private version (i.e., AugFLAIM (Private)). Furthermore, we find that some combination of (Private Hetero + Combine) gives the second best performance followed by (Non-private Hetero). This explains the discrepancy in accuracy between AugFLAIM (Private) and AugFLAIM (Non-private) in Section 5. The leading cause of which, is the estimation of 1-way marginals at every round, which gives a significant boost in utility. When combined with heterogenity measures, methods obtain their best utility overall. Our conclusions are similar on Adult, where there is not much difference in utility between the fully private and non-private methods when they both estimate 1-ways at each round.

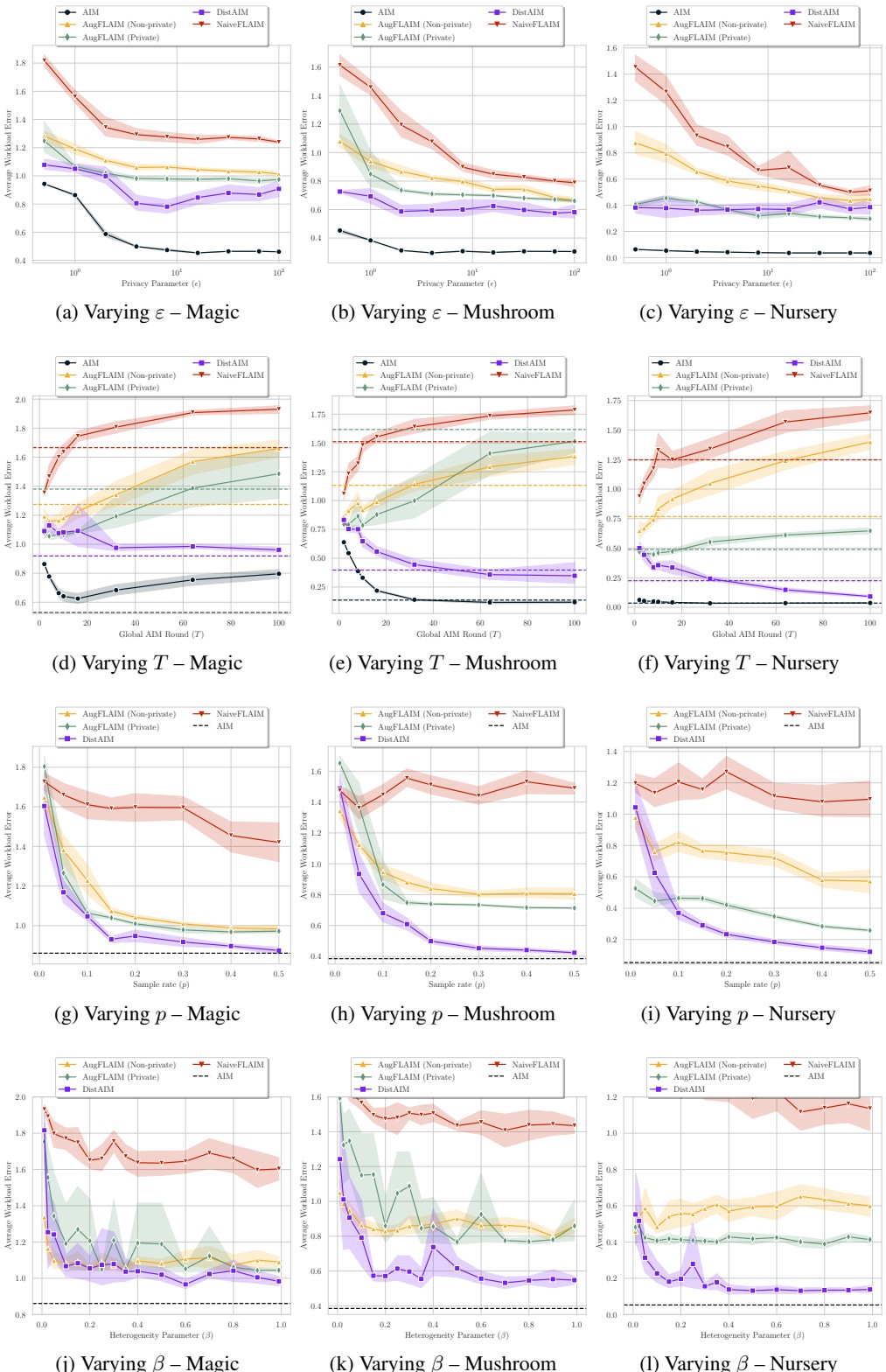

Figure 7: Repeated experiments from Section 5 but on the other datasets: Magic, Mushroom, Nursery

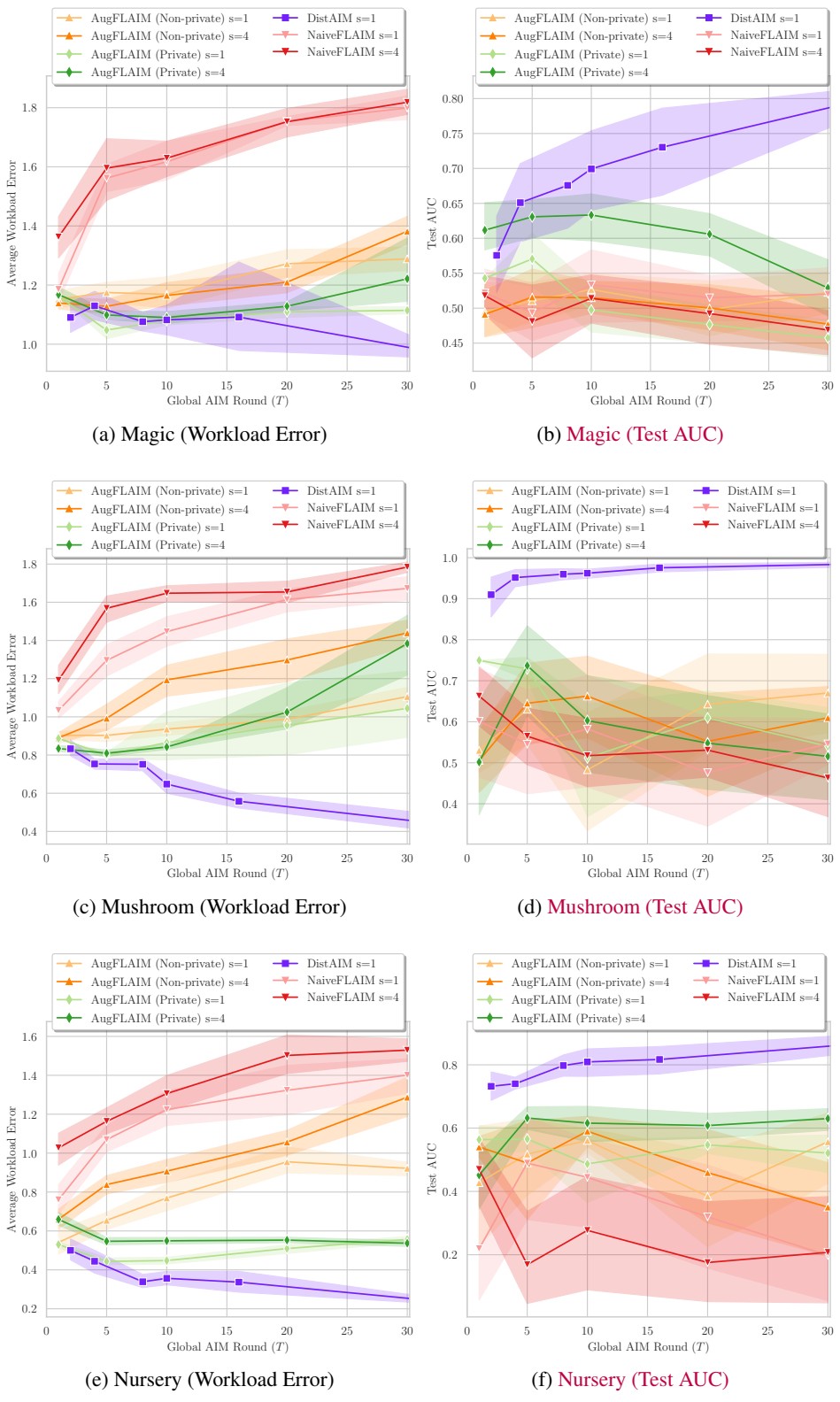

Figure 8: Varying local rounds $s \in \{1, 4\}$ as in Figure 2e but on alternative datasets.

