# OpenReview forum: "FLAIM: AIM-based Synthetic Data Generation in the Federated Setting"
_ICLR.cc/2024/Conference — Submitted to ICLR 2024_

### Official Review · Reviewer_DXga · 2023-10-24

**Soundness:** 2 fair
**Presentation:** 2 fair
**Contribution:** 2 fair
**Rating:** 3
**Confidence:** 3

**Summary:**

This paper studies differentially private data synthesis in the horizontal federated learning setting. The authors identify the key challenge of this problem is the data heterogeneity. In this paper, the authors propose two variants of the central AIM algorithm, DistAIM and FLAIM. Compared with DistAIM, the FLAIM is expected to rely on a more light-weight secure aggregation algorithm. The authors show the proposed algorithm can outperform the naive implementation of AIM in a federated learning setting.

**Strengths:**

1. The authors identify the challenges in differentially private data synthesis with heterogeneous local data in the federated learning setting.
2. The authors propose two different algorithms for solving the challenge of differentially private data synthesis with heterogeneous data.
3. The proposed FLAIM solution on how to handle the heterogeneity in marginal selection is novel.

**Weaknesses:**

* Some key elements of the algorithm are not clearly motivated or explained, leaving the effectiveness of the algorithm unjustified.
* Although it is acceptable that the DP data synthesis paper cannot provide a theoretical guarantee, some counter-intuitive phenomena in the experiments are not clearly explained.
* The writing needs to be improved.
  - Speaking at the paper structure level, while the core idea of the paper should be relatively straightforward, the paper's organization may introduce extra difficulties for readers to catch those ideas. Especially while the DistAIM and the proposed FLAIM are in the same section, it is not clear whether DistAIM is used as a motivation for FLAIM or serves as other purposes.
  - As for the notation level, the paper user both $u(q; D)$ and $u(D; q)$ for the EM utility score. In the algorithm, $\sigma_i$ (in line 14) may not be clearly defined (not sure whether it is used with $\sigma_t$ interchangeably).

**Questions:**

1. Why does the local client still need to "estimate the new local model via PGM"? This step looks strange because the local models are not aggregated globally, but it may affect the query selection and measurement in the following local rounds, making it unclear what local measurement error will be aggregated to the server.
2. What is the $\tilde{N}$ and $\sigma_i$ in line 14 of Algorithm 1?
3. Is there any theoretical performance guarantee for the algorithm?
4. Why is secure aggregation not applicable to the DistAIM?
5. What assumption of trust between participants and compute servers is relaxed when switching from DistAIM to FLAIM? How much overhead is reduced because of cryptographic protocol changes?
6. Why does AugFlaim (non-private) have worse performance than AugFlaim (private)? Does it mean too accurate heterogeneity information hurts the algorithm's performance?
7. Why there is no AugFlaim (non-private) results in Figure 2f?

---

> ### Author Response · Authors · 2023-11-17
> **Response to Reviewer DXga (1/2)**
>
> We would like to thank the reviewer for their detailed comments and would like to address their questions and concerns.
>
> > The writing needs to be improved.
>
> We have addressed the two points made about the paper structure and writing highlighted in your review (see below). Additionally, we have provided clearer conclusions in our experimental section and in the description of the FLAIM algorithm which has been highlighted in the new version of the paper.
>
> > Speaking at the paper structure level, while the core idea of the paper should be relatively straightforward, the paper's organization may introduce extra difficulties for readers to catch those ideas. Especially while the DistAIM and the proposed FLAIM are in the same section, it is not clear whether DistAIM is used as a motivation for FLAIM or serves as other purposes.
>
> The purpose of DistAIM is twofold. First, it acts as motivation for FLAIM since it is one way to perform synthetic data generation in the federated setting via AIM. However, the key motivation that follows is that DistAIM is not defined within the standard FL framework, and because of this can suffer from high overhead with its cryptographic protocols.
>
> Secondly, DistAIM acts as a baseline derived from the extant literature. It is a modification of the Pereira et al. work which has been adapted to use AIM and to work in the FL setting with client subsampling. Following your comment, we have modified the paper to include clearer motivation of FLAIM (via DistAIM) and to separate the sections.
>
> >As for the notation level, the paper user both U(q;D) and U(D;q)  for the EM utility score [...]
>
> We have fixed this inconsistency in the new version to use only $u(q;D$).
>
> >Why does the local client still need to "estimate the new local model via PGM"? This step looks strange because the local models are not aggregated globally, but it may affect the query selection and measurement in the following local rounds, making it unclear what local measurement error will be aggregated to the server.
>
> If the local client is performing a single local step, they do not need to estimate the new local model via PGM. They simply choose a marginal to submit based on the global model estimates that are received at the start of the round. Line 9 of Algorithm 1 can be omitted when local steps s=1. We have updated the algorithm to make this clearer.
>
> A point of interest with FLAIM is whether performing multiple local steps gives any significant utility increase. There is a natural analog with the standard training of federated neural networks, where clients train a local model for a number of local epochs (or local steps). However, unlike standard DP-FL training of NNs, the privacy cost of FLAIM training is not decoupled from the number of local rounds performed. In other words, you must scale the privacy cost in the number of local rounds (in addition to the number of global steps). In the case of s > 1 local steps, the local model does need to be updated since they are making multiple marginal selections
>
> > What is the N and sigma in line 14 of Algorithm 1?
>
> $\sigma$ is the associated standard deviation of the Gaussian noise that was added to the measurement. When the number of global rounds (T) is fixed, this is a constant for all measurements. If instead budget annealing is used, then sigma changes on a per-round basis (it is halved when the annealing condition passes, see Appendix A.1). In the presentation of Line 14 we have updated $\sigma_i$ to be $\sigma_t$ to make this clearer.
>
> N is the total number of samples that contributed to the measurement of a marginal across clients who provided it. To improve the clarity, we have updated the notation in the algorithm and defined it clearly at the end of Section 4.3 with a more detailed description.
>
> > Is there any theoretical performance guarantee for the algorithm?
>
> There are no theoretical utility guarantees for the algorithm in the federated setting. We note that there are also no theoretical utility guarantees of the AIM method in the central setting. It may be possible to provide some guarantee showing the utility of FLAIM is close to that of central AIM and is something we regard as future work.
>
> > Why is secure aggregation not applicable to the DistAIM?
>
> Secure aggregation is applicable only to the aggregation of marginals in the AIM algorithm. This means secure-aggregation could be used to share the workload answers but further cryptographic techniques must be used by DistAIM to compute the exponential mechanism which is required for choosing a marginal to add to the model at a given round. One reason for introducing FLAIM is that secure-aggregation can be used in a straightforward manner without further cryptographic protocols.

---

> > ### Author Response · Authors · 2023-11-17
> > **Response to Reviewer DXga (2/2)**
> >
> > > What assumption of trust between participants and compute servers is relaxed when switching from DistAIM to FLAIM? How much overhead is reduced because of cryptographic protocol changes?
> >
> > The trust between participants and compute servers for FLAIM depends on where the noise is added. We assume that participants trust the (single) compute server to add noise to obtain the differential privacy guarantee. For DistAIM, we assume a 3-party setting requiring an honest majority between compute servers. We note that the framework for DistAIM (which follows that of Pereira et al.) is flexible in its trust model and alternatives exist e.g. 2-party dishonest that require minor changes to the underlying cryptographic protocols.
> >
> > Regarding the overhead, we study only the total communication for clients participating in the training of DistAIM/FLAIM. For DistAIM, clients must secret-share all of their workload answers in each round they participate. For FLAIM, clients send simply s (where s is the number of local rounds) marginals to the server (under SecAgg). We assume that in either case, the secret-sharing follows the same cryptographic protocol so this does not affect our overhead comparisons.
> >
> > > Why does AugFlaim (non-private) have worse performance than AugFlaim (private)? Does it mean too accurate heterogeneity information hurts the algorithm's performance?
> >
> > To explicitly state the differences between AugFLAIM (Private) and AugFLAIM (non-private) in our experiments it helps first to break down the various modifications to NaiveFLAIM in the following way:
> >
> > *  Private hetero - Use the private heterogeneity measurement introduced in Section 4.3 (of the newly revised paper)
> > * Non-private hetero - Use the true heterogeneity measurement introduced in Section 4.2
> > * Combine - Estimate all 1-way marginals at every round (under DP) and feed these back to the PGM model. In the case of (Private hetero + combine) this is obtained without additional privacy cost. When used with (Non-private + combine) privacy budget is spent on estimating 1-ways at each round but the true heterogeneity measurement is used in utility scores.
> > * Filter - remove 1-way marginals from the workload. In the case of (filter + combine) this is done to prevent clients from measuring 1-way marginals multiple times in a round.
> >
> > Hence, in our experiments AugFLAIM (Private) = (Private hetero + combine + filter) whereas AugFLAIM (Non-private) = (Non-private hetero) with no combining or filtering. One of the key reasons for the utility improvement in AugFLAIM (Private) is that one-way marginals are repeatedly estimated during training. This seems to lower the overall workload error, possibly forcing the model to produce answers to higher order marginals that are more consistent with the 1-ways.
> >
> > We have now added an ablation to Appendix C that shows the effects of AugFLAIM with and without the various modifications described above on the Adult and Nursery datasets (Figure 6). Take for example the experiment on the Nursery dataset (Figure 6b). We observe that when we add the filter and combine modifications to the AugFLAIM variation that uses a non-private heterogeneity measure, it achieves performance better than the fully private version. We chose to omit this combination from the main experiments to avoid conflation between the effect of the true heterogeneity measure on utility vs. utility gained from repeatedly measuring 1-ways and in general it seems the combination of both gives best performance overall.
> >
> >
> > > Why there is no AugFlaim (non-private) results in Figure 2f?
> >
> > AugFLAIM (non-private) was removed from Figure 2f to reduce visual clutter. The finding for AugFLAIM (Non-private) is the same as that of AugFLAIM (Private) - we see a similar test AUC increase as we increase the number of local updates. We have updated the plots to include AugFLAIM (non-private) (Figure 2f, and Figures 8(b,d,f)).

---

### Official Review · Reviewer_B23T · 2023-10-28

**Soundness:** 3 good
**Presentation:** 2 fair
**Contribution:** 1 poor
**Rating:** 3
**Confidence:** 5

**Summary:**

This paper proposes a Federated Learning-based synthetic data generation method called FLAIM, a variation of the AIM algorithm, where data is distributed across multiple clients. The objective is to maintain individual privacy while collaboratively facilitating data sharing. FLAIM modifies AIM to handle heterogeneity and reduces overhead compared to traditional Secure Multi-party Computation (SMC) techniques. The proposed approach is evaluated on benchmark datasets and compared to other state-of-the-art methods, demonstrating improved utility while reducing overhead. This paper offers valuable insights into the challenges and solutions related to SDGs in a federated setting. The FLAIM algorithm proposed in the paper shows the potential to create effective synthetic data while maintaining privacy. The empirical study emphasizes the significance of considering heterogeneity in Federated Learning and the trade-offs between privacy and utility performance.

**Strengths:**

1) This paper suggests a new method for generating synthetic data in a Federated Learning setting while addressing the challenges of heterogeneity in federated settings.

2) After conducting a comprehensive assessment of the FLAIM technique on standard datasets, the authors compared its performance with other cutting-edge techniques. The results showed that the FLAIM method offers better efficiency with reduced overhead.

**Weaknesses:**

1) It remains a challenge to determine whether the FLAIM method would retain its efficiency when applied to real-world datasets that display more intricate structures and distributions, as its performance has been evaluated solely on benchmark datasets.

2) Although the paper compares the FLAIM method to other advanced methods, it does not give a complete comparison to all the related methods in the literature.

**Questions:**

I saw that you achieved significant performance improvement in the FL setting. What are the problems you will solve to re-implement the AIM algorithm in the FL setting?

---

> ### Author Response · Authors · 2023-11-17
> **Response to Reviewer B23T**
>
> We would like to thank the reviewer for their comments.
>
> > It remains a challenge to determine whether the FLAIM method would retain its efficiency when applied to real-world datasets that display more intricate structures and distributions, as its performance has been evaluated solely on benchmark datasets.
>
> We have selected a number of benchmark tabular datasets to be consistent with experiments performed in prior work in the central setting of DP. For example, the original AIM paper uses the Adult dataset in their experiments. Furthermore, we simulate the federation of these datasets using standard methods in the literature such as the label-skew approach introduced in [1].
>
> We believe these benchmarks are representative of ‘real-world’ data, which justifies their widespread adoption in the ML community.  Nevertheless, it is relevant to perform more evaluations on diverse data sets for FLAIM (as well as AIM). It is still challenging to run such tests in the federated setting because there are not “real-world” federated datasets freely available for tabular data. Benchmarks such as LEAF [2], aim to simulate realistic non-IID partitions but are for image/text data which is not well-suited for our task.
>
> [1] Li, Qinbin, et al. "Federated learning on non-iid data silos: An experimental study." 2022 IEEE 38th International Conference on Data Engineering (ICDE). IEEE, 2022.
>
> [2] Caldas, Sebastian, et al. "Leaf: A benchmark for federated settings." arXiv preprint arXiv:1812.01097 (2018).
>
> > ... it does not give a complete comparison to all the related methods in the literature.
>
> We performed a thorough literature review, and reported the most relevant examples in the paper. For the federated case, we are not aware of any other methods except for the work of Pereira et al. in distributing MWEM which we have compared to by using DistAIM which is an improved version. The closest alternative is to compare with deep learning synthesizers such as GANs. These can be federated privately by training within standard DP-FL frameworks i.e., using DP-FedSGD. However, many recent studies such as [3,4,5] have highlighted the performance gap between graphical models and deep learning synthesizers when trained with DP in the central setting. They show graphical model approaches consistently perform better on tabular datasets. In particular, iterative methods that use PGM rank high on average. Additionally, the performance gap is likely worsened in FL where approaches like DP-GAN will not scale as well.
>
> We already ran initial experiments to compare with alternative central baselines such as Bayesian methods (e.g. PrivBayes) and DP-GANs. We found that such methods perform worse than AIM on our benchmarks and even sometimes DistAIM/FLAIM. Hence, if we were to federate these methods they would follow the same performance gap (and likely be even worse).
>
> [3] Tao, Yuchao, et al. "Benchmarking differentially private synthetic data generation algorithms." arXiv preprint arXiv:2112.09238 (2021).
>
> [4] Liu, Yucong, Chi-Hua Wang, and Guang Cheng. "On the Utility Recovery Incapability of Neural Net-based Differential Private Tabular Training Data Synthesizer under Privacy Deregulation." arXiv e-prints (2022): arXiv-2211.
>
> [5] Ganev, Georgi, Kai Xu, and Emiliano De Cristofaro. "Understanding how Differentially Private Generative Models Spend their Privacy Budget." arXiv preprint arXiv:2305.10994(2023).
>
> > I saw that you achieved significant performance improvement in the FL setting. What are the problems you will solve to re-implement the AIM algorithm in the FL setting
>
> To recap, in Section 3 (of the revised paper), we introduce DistAIM, an extension of the recent work by Pereira et al. which replaces the poor in utility MWEM method with the SOTA method AIM. We change the setting by assuming only a proportion of clients are available to participate at a particular round which is common in practical FL scenarios. While this approach can have good utility it also requires significant overhead due to cryptographic protocols. We instead explore a more “traditional” approach in FL which is based on clients performing a number of local steps before sending back model updates. This leads us to develop NaiveFLAIM which is a natural analog to standard FL training. However, we highlight in Section 4.1 the key problem we have to solve when working with FLAIM. If we let clients make decisions locally (and across multiple local steps) they are typically biased by heterogeneity in their local datasets. In order to correct this, we introduce (in Section 4.2 and 4.3)  a non-private and private measure of heterogeneity that can be used to correct these issues and maintain utility. Furthermore, when working under the FLAIM framework we can avoid the use of heavyweight cryptography which is needed in DistAIM which results in smaller overheads when datasets have features with large cardinality.

---

### Official Review · Reviewer_gW6i · 2023-10-30

**Soundness:** 3 good
**Presentation:** 3 good
**Contribution:** 2 fair
**Rating:** 8
**Confidence:** 3

**Summary:**

This paper considers the problem of federated differentially private (DP) synthetic data generation (SDG). They start from the state of the art method AIM for DP SDG in the central model and consider multiple ways of distributing it. Firstly they consider a version of it implemented in secure multi-party computation (SMC), though they allow only a fraction of the data holders to be present at each step, introducing some extra error compared to using AIM. They then try to remove most of the heavy SMC by switching to a method based on federated learning, which introduces some more error from heterogeneity in the dataset. They then largely mitigate this new error using a private estimate of the heterogeneity to improve client choices.

They also provide an experimental section that shows that they do indeed get accuracy improvements from the parts of the algorithm designed to improve accuracy on various datasets.

**Strengths:**

The paper is clear and well written.
The results all seem reasonable and correct.
The privacy guarantees are rigorous.

**Weaknesses:**

The biggest question mark here is whether DP-SDG isgoing to be the practical answer in any situation, though this seems worth exploring anyway.

**Questions:**

Is the utility of the generated data actually good enough to make this a practical solution for any application?

---

> ### Author Response · Authors · 2023-11-17
> **Response to Reviewer gW6i**
>
> We would like to thank the reviewer for their feedback.
>
> >The biggest question mark here is whether DP-SDG is going to be the practical answer in any situation, though this seems worth exploring anyway.
>
> >Is the utility of the generated data actually good enough to make this a practical solution for any application?
>
> These are important points and are the core motivation for our work. Firstly, we note that FLAIM is not meant to be a practical solution to all applications. SDG methods using graphical models often perform poorly on high-dimensional datasets (in the central setting) and our FLAIM method will inherit these problems as well. Furthermore, FLAIM is limited to tabular generation and so is unsuitable for image/text.
>
> We claim that FLAIM does exhibit sufficient utility to allow it to be used for downstream tasks. Our primary notion of utility is the test AUC of classifiers trained on the synthetic data, as an exemplar task.  Figure 2f in our paper shows our best FLAIM methods can consistently achieve a test AUC of ~0.8 for $\varepsilon=1$ on the Adult dataset. This is not far from the central AIM result of ~0.85 and would be a practically useful classifier. In our experiments, we often present the average L1 workload error to be consistent with prior work in this area. Here, there is an appreciable gap between the federated and central workload errors. However, this is not always predictive of the test AUC as we have shown.

---

### Author Response · Authors · 2023-11-17
**Response to all Reviewers**

We would like to thank all reviewers for their comments and feedback. We have uploaded a new version of the paper taking into account the concerns raised. Any changes to the paper are highlighted in dark red.

We will post separate comments to each of the reviewers addressing their individual questions.

---

### Meta-Review · Area_Chair_dEs8 · 2023-12-07

**Metareview:**

While there was some support for the paper, overall, they were not strong enough. Furthermore, I took a closer look at the paper. It seems the paper feels like a stitching ideas from multiple different papers, without a single idea that stands out. The only challenge I found that was non-trivial for lifting the existing literature for synthetic data generation to the distributed case is the implementation of exponential mechanism. If I understand correctly, even that problem is solved via the prior work of Pererira et al 2022 (which the authors of the current paper do clearly acknowledge). So, the recommendation would be to update the paper with clear details about the contribution/ key new-ideas.

**Justification For Why Not Higher Score:**

The technical contribution seemed marginal.

**Justification For Why Not Lower Score:**

NA

---

### Decision · Program_Chairs · 2024-01-16

Reject